# Rethinking Channel Dimensions to Isolate Outliers for low-bit weight quantization of Large Language Models

**Jung Hwan Heo** [*][†][1] **Jeonghoon Kim** [*][‡][2] **Beomseok Kwon**[2]
**Byeongwook Kim**[2] **Se Jung Kwon**[2] **Dongsoo Lee**[2]
[1] University of Southern California
[2] NAVER Cloud

## Abstract

Large Language Models (LLMs) have recently demonstrated remarkable success across various tasks. However, efficiently serving LLMs has been a challenge due to the large memory bottleneck, specifically in small batch inference settings (e.g. mobile devices). Weight-only quantization can be a promising approach, but sub-4 bit quantization remains a challenge due to large-magnitude activation outliers. To mitigate the undesirable outlier effect, we first propose per-IC quantization, a simple yet effective method that creates quantization groups within each input channel (IC) rather than the conventional per-output-channel (per-OC). Our method is motivated by the observation that activation outliers affect the input dimension of the weight matrix, so similarly grouping the weights in the IC direction can *isolate outliers within a group*. We also find that activation outliers do not dictate quantization difficulty, and inherent weight sensitivities also exist. With per-IC quantization as a new outlier-friendly scheme, we propose **Ada**ptive **Dim**ensions (**AdaDim**), a versatile quantization framework that can adapt to various weight sensitivity patterns. We demonstrate the effectiveness of AdaDim by augmenting prior methods such as Round-To-Nearest and GPTQ, showing significant improvements across various language modeling benchmarks for both base (up to $+4.7\%$ on MMLU) and instruction-tuned (up to $+10\%$ on HumanEval) LLMs. Code is available at: https://github.com/johnheo/adadim-llm

## 1 Introduction

The rise of Transformers (Vaswani et al., 2017) has led a remarkable success of Large Language Models (LLMs) (Brown et al., 2020; Touvron et al., 2023), achieving on par or excelling human-level performance on various tasks (Bubeck et al., 2023). However, with the rapid scaling in model size, efficiently serving LLMs has become a significant challenge. Specifically, the autoregressive decoding of an LLM is limited by memory bandwidth rather than compute (Kim et al., 2023b).

Low-bit weight quantization is a promising approach to reduce storage and accelerate inference latency (Park et al., 2022). By reducing weight precision, one can pack multiple weights under equal bit width to increase memory I/O (e.g., $4\times$ for FP16 $\rightarrow$ INT4). However, sub-4 bit quantization remains a challenge due to the presence of *activation outliers* in billion parameter scale modern LLMs (Dettmers et al., 2022; Bondarenko et al., 2023). Prior works have sought to mitigate large activations amplifying the rounding errors in the corresponding weights, where a small subset of weights is much more sensitive than others (Lin et al., 2023; Dettmers et al., 2023; Yuan et al., 2023).

In this paper, we first propose per-input-channel (per-IC) quantization, a simple yet effective scheme to address the activation outlier problem. The motivation behind per-IC quantization lies in how activation outliers impact specific input channels of the weight matrix. Thus, similarly grouping the weights in the IC direction can effectively isolate the effect of these outliers (ref. Figure 1). Such an approach is particularly feasible in the weight-only quantization context because it does not rely on specialized INT8 GEMM kernels impose the per-OC grouping constraint.

---

[*]Equal contribution
[†]Work done during an internship at NAVER Cloud
[‡]Corresponding author:jeonghoon.samuel@gmail.com

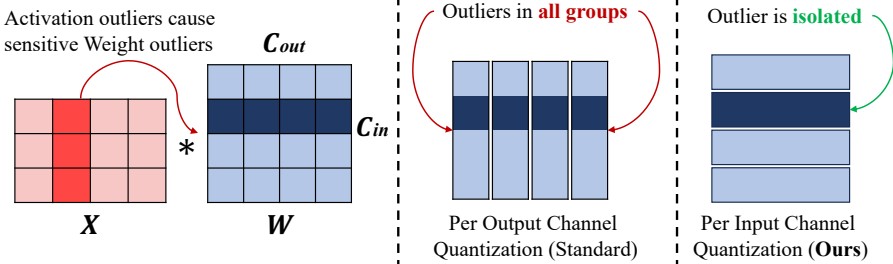

Figure 1: **Per-input-channel quantization.** Activation outliers that affect certain input channels (ICs) amplify quantization errors. Such sensitive ICs exist in *all groups* in the conventional per-output-channel (per-OC) quantization. With our per-IC scheme, the outlier effect is *isolated*.

By *unlocking **input dimension** as a new design parameter* that helps to sidestep the activation outlier problem, we propose AdaDim, a versatile quantization framework that can adapt to different weight sensitivity scenarios. We now organize our contributions below:

- From analyzing the structural relationship between activation outliers and sensitive weights, we propose per-IC quantization. Unlike traditional per-OC quantization where the outlier effect is *pervasive*, per-IC quantization *isolates* the outlier effect.

- Activation outliers emerge only in a subset of the network, prompting a *selective* application of per-IC quantization. We present Adaptive Dimensions (AdaDim), a framework that adapts to various weight sensitivity patterns by using both per-IC and per-OC quantization.

- Augmenting weight-only quantization methods such as Round-To-Nearest (RTN) and GPTQ with AdaDim results in significant boost in various language modeling abilities and specialized tasks (math, coding) for both base and instruction-tuned LLMs.

## 2 RELATED WORK

Generative inference of an LLM is heavily memory bound (Sheng et al., 2023). In a single-batch inference setting which is usually common for mobile devices, *billions* of floating point weights need to be read in from VRAM into on-chip cache to generate a *single* next token. Thus, smaller weight precision can make memory I/O more efficient by packing multiple values under equal bit width.

### 2.1 LLM QUANTIZATION

Quantization is an effective method to reduce weight precision, accelerating inference and reducing storage. INT8 quantization maps both activations and weights to lower precision, so that specialized GEMM engines can effectively accelerate arithmetic computation for large matrix mulplications (Xiao et al., 2022). Thus in autoregressive decoding workloads as in modern LLMs (Touvron et al., 2023), INT8 quantization is helpful for large batch sizes where compute is the bottleneck, but not for small batches (let alone a single batch) where memory is the bottleneck.

An alternative to address the memory bottleneck is weight-only quantization (Park et al., 2022), which leaves activations in high precision (e.g., FP16) while pushing the weights to even lower precision ($\leq 4$ bits). We focus on weight-only quantization to accelerate memory I/O rather than compute, as small-batch inputs do not saturate the powerful compute capacity of modern GPUs(Kim et al., 2023b). In order to preserve accuracy while minimizing the number of bits, group-wise per-channel quantization is commonly used (Shen et al., 2020; Kim et al., 2023a). It is a fine-grained quantization scheme where a group of consecutive weights inside each channel share the same quantization parameters.

### 2.2 THE ACTIVATION OUTLIER PROBLEM

**INT8 quantization.** Low-bit transformer quantization is complicated by the presence of activation outliers (Bondarenko et al., 2023). First characterized in OPT models (Zhang et al., 2022) by Dettmers et al. (2022), activation outliers emerge in a small subset of hidden dimensions and have up to $20\times$

larger magnitude than other channels. Such outliers increase quantization range, which results in most of the activation values being mapped to the same quantization bucket, making quantization ineffective. To address this, recent works have attempted to suppress or smooth such outliers (Wei et al., 2022b; Xiao et al., 2022) or use training (Liu et al., 2023b; Lee et al., 2023) to achieve acceptable INT8 accuracy.

**Weight-only quantization.** Although activation outliers can be seemingly unrelated, it has been discovered that they also make weight quantization difficult (Dettmers et al., 2023; Lin et al., 2023). When large activations are multiplied by dequantized weights that inherently possess rounding errors, even small errors can be amplified by the large inputs, causing non-trivial perturbations to the layer output. To address the undesirable outlier effect, existing per-channel quantization methods keep outliers in higher precision via mixed-precision formats (Dettmers et al., 2023; Kim et al., 2023b) or use scaling to give higher importance during quantization (Lin et al., 2023).

**Reviewing renewed GPTQ.** A pioneering work in LLM weight-only quantization is GPTQ (Frantar et al., 2022) which does iterative per-channel quantization while compensating the rounding errors with Hessian-based approximation. Contrary to what is explained in the original GPTQ paper as *Arbitrary Order Insight*, it has been found that the order in which channels are quantized are actually crucial (i.e., --act-order option in their repo[1]). Because salient (sensitive) weight channels compensating the error of non-salient weight channels is undesirable, the activation reordering technique sorts the channels in the descending order of activation magnitude to quantize salient channels first (Figure 8). Although this provides better accuracy, reordering incurs non-trivial hardware overhead due to quantization scales becoming non-contiguous in memory (Lin et al., 2023).

To avoid inefficient random memory access, GPTQ added another option (i.e., --static-groups) that pre-computes the quantization parameters *before* reordering. This enables activation-aware weight updates while quantization parameters are still contiguous in memory. However, we find that such static quantization constraint fails to capture the changing weight dynamics that occur *during* iterative weight updates, leading to suboptimal accuracy (Table 2b).

## 3 METHODOLOGY

**Overview.** In this section, we first structurally analyze the relationship between activation outliers and weight sensitivity patterns (Figure 2). We then introduce per-IC quantization to isolate the outlier effect, empirically validating its effectiveness in a preliminary study (Table 1). Finally, we propose Adaptive Dimensions (AdaDim), a versatile framework that can automatically choose whether to use per-IC or per-OC quantization for each layer of the network.

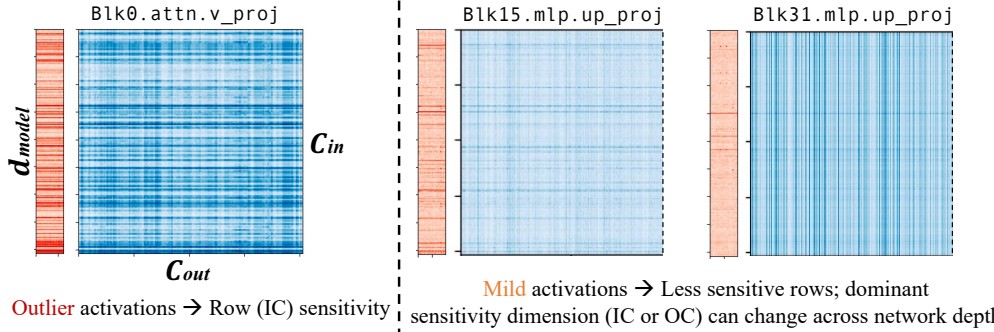

Figure 2: **Weight sensitivity patterns** of LLaMA-V2-7B. Darker colors indicate larger activation magnitude (red), and higher weight sensitivity (blue). Sensitivity is computed with fisher information approximated by the squared of the gradient. We downsample the grids with a 16x16 maxpool kernel and take the log scale for clarity. **Left:** The presence of activation outliers lead to sensitive rows. **Right:** Mild activations lead to less sensitive rows, while the dominant sensitivity dimension (row or column) can change across network depth even for the same module.

---

[1]https://github.com/IST-DASLab/gptq

Table 1: **Isolating outliers with per-IC quantization.** We observe the largest activations before `attn.qkv` and `mlp.down` projections for LLaMA-V2 base models. Selectively applying per-IC quantization to RTN *only where outliers are present* gives the most improvement in perplexity and multi-task in-context learning performance (MMLU) on INT4 with group size 128.

| Model Size | Metric | FP16 | Baseline (RTN, *Per-OC*) | Module to apply *Per-IC* quant. | | | |
|---|---|---|---|---|---|---|---|
| | | | | (1)`attn.qkv` | (2)`mlp.down` | (1)&(2) | All |
| **7B** | Wiki-2 ppl. ($\downarrow$) | 8.79 | 9.22 | 9.17 | 9.11 | **9.09** | 9.11 |
| | MMLU 5-shot ($\uparrow$) | 45.98 | 44.54 | 44.77 | 44.7 | **45.21** | 44.38 |
| **13B** | Wiki-2 ppl. ($\downarrow$) | 7.89 | 8.13 | 8.12 | 8.11 | **8.10** | 8.13 |
| | MMLU 5-shot ($\uparrow$) | 55.61 | 54.43 | 54.76 | 54.90 | **54.97** | 54.67 |

## 3.1 ANALYZING WEIGHT SENSITIVITY PATTERNS

Although activation outliers are prevalent in modern LLMs (Wei et al., 2022c; 2023), they do not occur in every layer of the network. We investigate the structural relationship between activation outliers and sensitive weight outliers in the LLaMA-V2 base model family. We define weight sensitivity by using fisher information following Kim et al. (2023b), which can be approximated by the squared of the gradient obtained by using a calibration set. Here, sensitivity is defined per weight instead of per channel, as we infer channel sensitivity from the sensitivity of its constituent individual weights.

Our preliminary study shows that the largest activations occur before QKV attention projection and DOWN feedforward projection (Figure 9). When visualizing the sensitivity of the corresponding weight matrices, we find that the hidden dimensions where activation outliers emerge have high correlation to sensitive weight channels (left of Figure 2). However, even when outliers exist before Q and K projections, they can exhibit dominant OC sensitivity (Figure 10a). Thus, activation outliers cause sensitive rows but does not necessarily dictate the overall sensitivity.

Besides, when activation outliers do not exist, we observe that the weight matrix can have a mixture of sensitive IC and OC channels. Moreover, the dominant sensitivity dimension can actually switch across network depth, even if it is the same module (right of Figure 2). These observations motivate a weight quantization method that can *adapt to different sensitivity scenarios*, which is largely conditioned by the existence of activation outliers.

## 3.2 PER-IC QUANTIZATION

**Motivation.** One common thread of existing per-channel quantization methods is their usage of per-OC channel quantization. When activation outliers emerge in certain hidden dimensions, the amplification effect is permeated across all quantization groups for per-OC quantization (Figure 1). In contrast, grouping within each IC yields a 1:1 mapping between hidden dimension to a quantization group which isolates the outlier effect to be within a group. Thus, per-IC quantization can be a more effective method that mitigates the outlier problem.

**Selective outlier isolation.** To first verify our intuition that per-IC quantization can isolate the undesirable activation outlier effect, we augment the standard RTN method. As in Table 1, utilizing per-IC quantization for modules influenced by activation outliers can effectively improve both perplexity and multi-task in-context learning ability of an LLM. We also observe that *selective* utilization per-IC quantization is important; naively applying it to all modules can actually hurt baseline RTN performance from 44.54 to 44.38, while selectively applying it to QKV and DOWN modules improve up to **0.67%** on average MMLU score. Thus, it is desirable to search for a *selective* scheme to apply per-IC quantization, motivating our adaptive approach in Section 3.3.

## 3.3 ADAPTIVE PER-CHANNEL QUANTIZATION

**Optimization objective.** Beyond heuristically determining the channel quantization dimension by looking at the sensitivity patterns offline, we propose an adaptive method that can achieve this on the fly during quantization. For each linear layer $\mathbf{W}_\ell$ (linear projection $\mathbf{W}$ at layer $\ell$) in a neural network, we formulate channel quantization as a simple binary selection problem that chooses the optimization parameter **dim** as either one of the two (OC or IC) dimensions. To measure which dimension is more effective, we adopt the widely used reconstruction error metric (Nagel et al., 2020; Li et al., 2021), yielding the optimization objective as

$$\mathbf{dim}^* = \underset{\mathbf{dim}\in\{oc,ic\}}{\arg\min} \mathcal{L}(\mathbf{dim}), \quad \mathcal{L}(\mathbf{dim}) = \|Q_{\mathbf{dim}}(\mathbf{W}_\ell)\mathbf{X}_\ell - \mathbf{W}_\ell\mathbf{X}_\ell\|, \tag{1}$$

where the reconstruction error $\mathcal{L}(\mathbf{dim})$ is defined as the L2 distance between the outputs of a full precision layer $\mathbf{W}_\ell\mathbf{X}_\ell$ and that of the quantized layer $Q(\mathbf{W}_\ell)\mathbf{X}_\ell$. To obtain $\mathbf{X}$, we curate a small calibration set by randomly sampling from the pretraining corpus (e.g. The Pile). Here, the per-channel quantization function $Q_{\mathbf{dim}}$ can either create quantization groups per-OC (standard) or per-IC (proposed), as illustrated in Figure 1. Since the search space of the dimension parameter is only two, AdaDim requires a very small number of forward passes to determine the optimal dimension.

**Augmenting RTN and GPTQ.** Applying our AdaDim to RTN is straightforward: we independently quantize the full precision weights two different times (per-IC and per-OC), then choose the dimension that yields a lower reconstruction error. Hence, we always search for the optimal dimension with RTN and optionally apply GPTQ in the selected dimension. As the GPTQ (Frantar et al., 2022) algorithm consists of iteratively 1) computing the quantization error of a weight channel and 2) applying hessian-based weight updates, using our per-IC variant simply requires executing the quantization step 1) with per-IC RTN. We also considered applying AdaDim to AWQ but found it incompatible (see Appendix A.2 for further discussion). Furthermore, per-IC GPTQ can obtain additional benefits from the reordering scheme, which we detail further in Appendix A.1.

## 4 EXPERIMENTS

### 4.1 EVALUATION SETUP

**Quantization setting.** In this work, we focus on weight-only per-channel (w/ uniform asymmetric setting) quantization with group size of 128, which is shown to be a good accuracy/latency trade-off point (Dettmers & Zettlemoyer, 2022). We also focus on INT3 quantization since it shows the biggest relative improvements while INT4 quantization yields comparable performance across methods (see Appendix C). Following the settings of GPTQ and AWQ, we use a small calibration set from the Pile (Gao et al., 2020) dataset. For instruction-tuned models, we also experiment with *task-specific* calibration sets, which is randomly sampled from the training split of the respective tasks.

**Models.** For base model evaluation, we use version 2 (V2) instead V1 of the LLaMA (Touvron et al., 2023) family with the exception of 33B since it is not yet publicly available. We further benchmark instruction-tuned models from the WizardLM (Xu et al., 2023) family, which is a series of performant LLaMA-V2 models fine-tuned with specialized instruction dataset curation. We use WizardMath and WizardCoder-Python (Luo et al., 2023a;b) to test mathematical reasoning and code generation ability.

**Tasks.** Following previous literature (Dettmers et al., 2022; Yao et al., 2022), we evaluate the quantized models on zero-shot commonsense reasoning (CSR) ability, including PIQA (Bisk et al., 2020), HellaSwag (Zellers et al., 2019), WinoGrande (Sakaguchi et al., 2019), and ARC-easy (Clark et al., 2018). Besides common sense abilities, we also evaluate multi-task generalization ability with five-shot setting (in-context learning) on MMLU (Hendrycks et al., 2020), which consists of 57 tasks including STEM, humanities, social science, and others (business, health, etc.). We used `lm-eval-harness` (Gao et al., 2021) for CSR and a reproducible repo[2] for MMLU. We report the average score of the four aforementioned CSR tasks and the total average of MMLU.

Following (Luo et al., 2023a;b), we evaluate instruction-tuned models on mathematical reasoning with Chain-of-Thought (CoT) prompting (Wei et al., 2022a) on GSM8k (Cobbe et al., 2021) dataset, a set of grade school math questions. We also test code generation with greedy decoding on the HumanEval (Chen et al., 2021) dataset, which includes hand-crafted programming problems written in Python. To curate domain-specific calibration sets, we use the training splits from GSM8k and from MBPP (Austin et al., 2021), a crowd-sourced entry level Python problem set.

**Baselines.** We benchmark against vanilla round-to-nearest quantization (RTN), GPTQ (Frantar et al., 2022), and AWQ (Lin et al., 2023) for LLM weight quantization. We thoroughly ablate GPTQ's most up-to-date design parameters as in Table 2 to find a strong baseline for modern LLMs such as LLaMA-V2 (Touvron et al., 2023). We first observe that heuristically fixing which module to apply per-IC quantization offline is inferior to the optimization-based (adaptive) setting where

---

[2]https://github.com/QwenLM/Qwen-7B/blob/main/eval

Table 2: **GPTQ ablation** using LLaMA-V2-13B with MMLU 5-shot average score. We see that adaptive method is superior to the heuristic-based approach. Although reordering by itself gives the best results for OC cfg., it is hardware inefficient. Thus, our final design is marked in green.

(a) **Heuristic-based.** Results when the decision to use per-IC quantization with reorder is fixed according to a heuristic from an offline observation as in Table 1.

| per-IC module | Accuracy (%) |
|---|---|
| none | 50.56 |
| attn.qkv | **51.47** |
| mlp.down | 50.4 |
| qkv & down | 48.93 |
| all | 51.4 |

(b) **Optimization-based.** Results when AdaDim adaptively chooses either per-OC/per-IC quantization for each layer by minimizing the reconstruction error.

| cfg. OC \ IC | default | reorder |
|---|---|---|
| default | 51.07 | 51.54 |
| reorder | 52.18 | 52.68 |
| reorder + static | 51.32 | **52.27** |

dimensions are chosen on-the-fly. Once the optimal dimension is selected with RTN, there are various combinations to apply the GPTQ algorithm: for OC dimension, we test default (baseline), reorder (activation reordering), and the reorder + static (hardware-efficient reordering) option. We do not test static groups for IC since it is already hardware-efficient. We empirically determine that static reordering for per-OC and reordering for per-IC yields the best accuracy-efficiency trade-off.

## 4.2 BASE MODEL QUANTIZATION

Base models serve as the fundamental backbone for modern LLMs, which demonstrated remarkable capabilities in general knowledge understanding (Bubeck et al., 2023). To test the effect of quantization on such models, we evaluate the LLaMA family that is widely used today (Taori et al., 2023; Liu et al., 2023a). We evaluate INT3 per-channel quantization with group size 128 (w3g128) on common sense reasoning with 0-shot and MMLU with 5-shot (in-context learning). As in Figure 3, AdaDim enables notable improvements in existing methods such as RTN and GPTQ. Remarkably, augmenting RTN with per-IC quantization yields a **4.7%** MMLU accuracy boost on the 7B model, surpassing both AWQ and GPTQ.

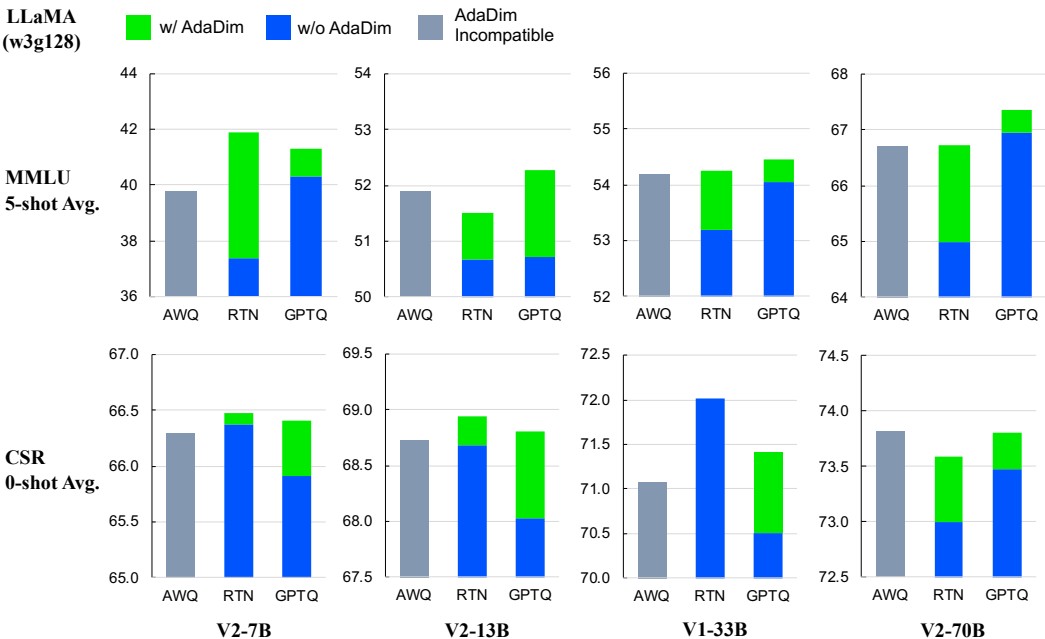

Figure 3: **Base model results.** Evaluating the effectiveness of our AdaDim framework for LLaMA base models on Massive Multi-task Language Understanding (MMLU) and commonsense reasoning (CSR) tasks. Performance boosts (in green) indicate additional gains from adaptively switching to per-IC quantization. We observe notable gains over the original per-OC versions of RTN and GPTQ (Frantar et al., 2022), often matching or even surpassing AWQ (Lin et al., 2023).

Table 3: We evaluate Vicuna, a family of instruction-tuned LLaMA models with improved language modeling. AdaDim applied to RTN and GPTQ are trailed by postfix '-ada' and highlighted in green.

| w3g128 | Vicuna-V1.5-7B | | Vicuna-V1.5-13B | | Vicuna-V1.3-33B | |
|---|---|---|---|---|---|---|
| | MMLU Avg. | CSR Avg. | MMLU Avg. | CSR Avg. | MMLU Avg. | CSR Avg. |
| FP16 | 50.27 | 67.45 | 56.02 | 69.81 | 59.22 | 71.12 |
| AWQ | 45.79 | 65.4 | 53.27 | **68.06** | 55.3 | 69.24 |
| RTN | 45.18 | 65.61 | 51.61 | 67.90 | 53.39 | 68.75 |
| RTN-ada | 46.25 | **66.42** | 51.8 | 67.92 | 55.93 | **69.42** |
| GPTQ | 47.32 | 64.88 | 53.11 | 67.55 | 55.68 | 68.08 |
| GPTQ-ada | **47.45** | 65.7 | **53.29** | 67.72 | **57.08** | 69.37 |

Table 4: **Task-specific quantization.** We evaluate instruction-tuned WizardLM models on math and coding. Intuitively, we find that using a calibration set from the target domain (e.g., code-code) rather than generic text corpus (e.g., pile-code) improves performance. Applying AdaDim (labeled green) consistently improves both RTN and GPTQ with up to 10% on HumanEval, surpassing AWQ.

| w3g128 | GSM8k pass@1 (↑) | | | | HumanEval pass@1 (↑) | | | |
|---|---|---|---|---|---|---|---|---|
| | WizMath-7B | | WizMath-13B | | WizCoder-Py-7B | | WizCoder-Py-13B | |
| FP16 | 55.35 | | 63.38 | | 55.49 | | 64.02 | |
| RTN | 32.52 | | 49.13 | | 35.37 | | 50.61 | |
| calib. set | base | target | base | target | base | target | base | target |
| AWQ | 39.42 | 40.49 | 55.19 | 54.97 | 43.29 | 44.51 | 57.32 | 60.36 |
| RTN-ada | 37.38 | 39.12 | 50.95 | 53.15 | 42.68 | 45.12 | **60.37** | 60.98 |
| GPTQ | 38.29 | 41.09 | 54.21 | 57.16 | 31.71 | 45.12 | 53.05 | 56.71 |
| GPTQ-ada | **41.77** | **42.15** | **56.78** | **57.47** | **46.34** | **46.95** | 53.69 | **62.2** |

## 4.3 INSTRUCTION-TUNED MODEL QUANTIZATION

Instruction tuning has become the method of choice to boost the performance and user interaction experience of LLMs (Wei et al., 2021; Sanh et al., 2021; Chung et al., 2022). A well-known instruction-tuned model that is also publicly available is Vicuna (Chiang et al., 2023) which is fine-tuned from LLaMA models. To benchmark how quantization affects their improved language modeling capabilities, we conduct MMLU and CSR evaluations in Table 3. AdaDim brings consistent improvements across various model scales; similar to the base model results, strong performances are displayed for RTN-ada on CSR and GPTQ-ada for MMLU. Performance boost is most noticeable in the 33B scale, where our adaptive solutions can bridge the degradation down to ∼2% points.

**Task-specific quantization.** Beyond general abilities like commonsense reasoning, fine-tuning to a specific task has shown to be effective in creating specialized LLMs (Luo et al., 2023a;b). WizardLM family is a set of LLaMA fine-tuned models that show very strong performance on open benchmarks such as GSM8k and HumanEval. Different tokens produce different activations; to better simulate the test-time activation distribution of a task-specific LLM, it may be desirable to also use a task-relevant calibration set. To test this intuition, we use both the generic text corpus (WikiText-2, denoted as *base*) and task-relevant corpora (GSM8k for math and MBPP for coding, denoted as *target*). As in Table 4, we confirm the proposition that using a target calibration set can bring improvements, sometimes significantly up to **8.51%** on the 13B coding model for GPTQ-ada. Notably, RTN-ada brings up to a **10.3%** boost over vanilla RTN, with GPTQ-ada outperforming all other methods when using the target calibration set which may be due to the task-specific weight updates that can serve as further fine-tuning.

## 4.4 ANALYSIS

**Sweeping precision ranges.** To test the generality of AdaDim across various quantization settings, we use LLaMA-V2-7B to sweep INT3/INT4 precision with 256, 128, and 64 group sizes. As in Figure 4, AdaDim strictly improves perplexity scores when applied to RTN and GPTQ. Prominently, RTN-ada shows significant perplexity improvements from vanilla RTN up to 1.06 on w3g256, even

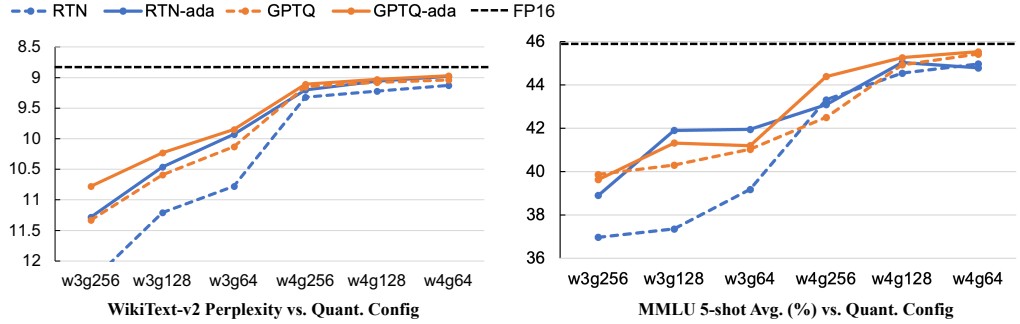

Figure 4: Sweeping various quantization configurations for LLaMA-V2-7B. Average bits per weight increases from left to right (x-axis). AdaDim can further close the gap between INT3/4 and FP16.

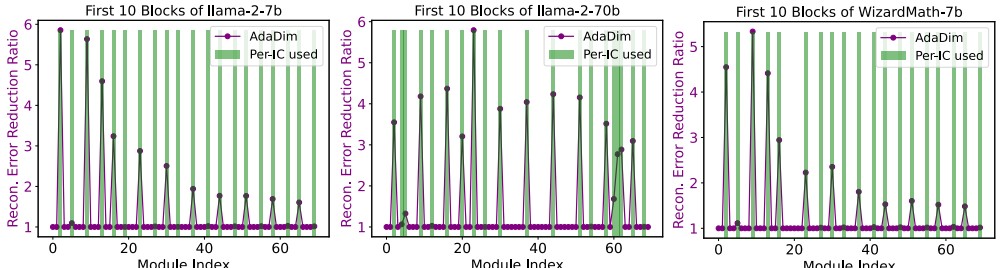

Figure 5: **Adaptive dimension selection.** By adaptively switching to per-IC quantization, AdaDim can reduce the reconstruction error up to $6\times$ in the RTN setting. The decisions vary across model size (7B vs. 70B) and task (language modeling vs. math), showcasing the versatility of AdaDim.

surpassing GPTQ on various ranges. On MMLU, AdaDim provides accuracy lifts that are relatively non-uniform, which we speculate is due to the nature of in-context learning.

**Reduced reconstruction error.** As in Figure 5, adaptively switching to per-IC quantization for RTN-ada can save up to $6\times$ in reconstruction error, aligning with our objective in Eq. 1. We notice that more savings come from earlier layers, which suggest that earlier part of the network is especially sensitive to activation outliers. Among the different modules, `attn.v` projection layer yields the most error reduction, followed by the `mlp.down` projection layer. The switch to per-IC decisions occur at different modules for different model sizes and tasks, demonstrating the versatility of AdaDim.

**Localized GPTQ updates.** Activation outlier effect is pervasive in the standard per-OC quantization, while it is isolated in the per-IC quantization (Figure 1). We study the impact of outlier isolation on GPTQ weight updates in Figure 6. GPTQ computes the weight update (analogous to a gradient) by first capturing the rounding error then multiplying it by the inverse hessian (formed by activations). Thus, quantizing the sensitive weights altogether with IC grouping will lead to larger quantization errors, but it will be localized to a few channels where the outliers emerge. Indeed, per-IC quantization uses larger, localized updates that target only a few input channels. In contrast, standard per-OC quantization uses smaller updates to much more channels, caused by the pervasively spread outlier effect. Hence, per-IC quantization minimally perturbs the weight distribution by focusing on only a few sensitive channels that contain densely populated outliers.

### 4.5 PER-IC KERNEL IMPLEMENTATION

We utilize the implementation of LUT-GEMM (Park et al., 2022), a Lookup Table-based (LUT-based) matrix multiplication method. LUT-GEMM first involves the precomputation stage, where possible multiplication outcomes between quantized weights and activations are stored in a Lookup Table. During actual computation in a forward pass, the matmul operations take the form of indexing the LUT (instead of multiplying) using the weight values as keys, enabling efficient GEMM operations. For our per-IC quantization, a similar implementation is applied. During LUT generation, each per-IC quantization scaling factor can be multiplied in advance with activations to create the Lookup Table. In cases of group quantization, adjusting tile sizes to fit the groups and generating LUTs on a

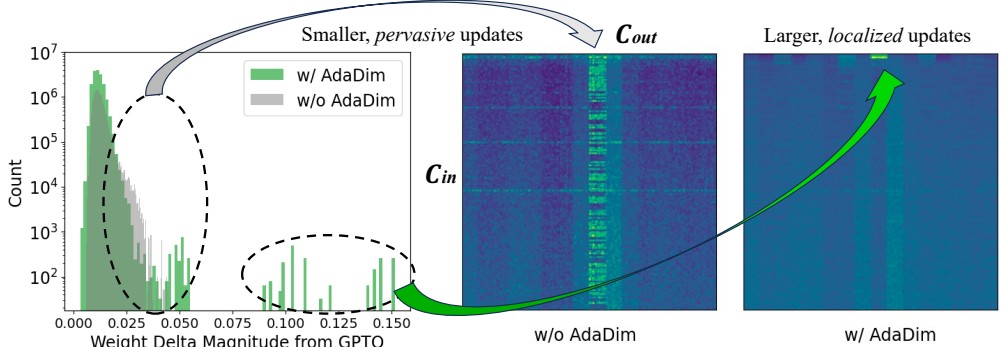

Figure 6: **GPTQ update behavior.** Thanks to per-IC quantization's outlier isolation, GPTQ's error compensation (weight update) is localized to a small subset of input channels instead of many input channels. By grouping sensitive weights together, AdaDim can use larger, localized updates that minimally perturb the original weight distribution. **Left:** A typical distribution of GPTQ's weight updates (magnitude) in LLaMA-V2-7B's `attn.v` layer. **Middle/Right:** Weight updates visualized in the 2D matrix form ($\mathbf{C_{in}}, \mathbf{C_{out}}$), zoomed in to the top right quadrant.

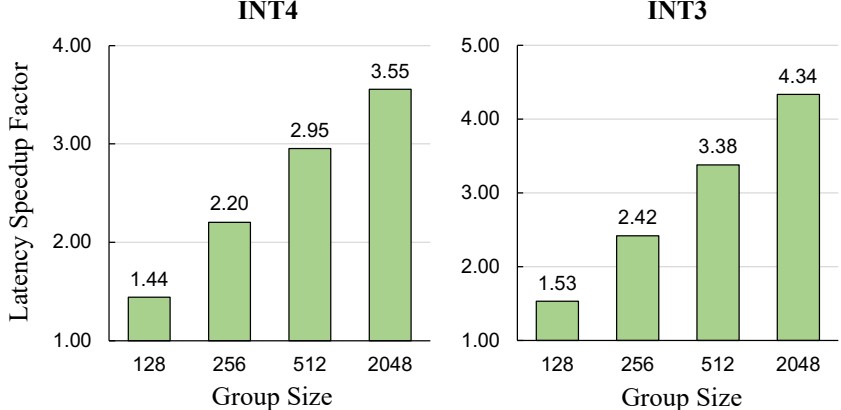

Figure 7: **Latency speedup of our Per-IC kernel over cuBLAS across various group sizes.** We measure the latency of the first FFN layer on OPT-175B model with 3-bit and 4-bit precision and corresponding kernel selections with hidden dimension size set to 12288 on an A100-80GB-GPU.

tile-by-tile basis allows for a similar application of this method. Due to time and resource limitations, we do not perform experiments with a fully optimized kernel that may further optimize the latency of our approach. Nevertheless, our per-IC kernel exhibits faster latency than the cuBLAS baseline as in Figure 7. This indicates that our Per-IC quantization not only achieves accuracy improvements but also leads to measurable speedups in inference latency. Further investigation into a well-optimized per-IC kernel is a critical direction for our future research. For further discussion, please refer to Appendix D.

## 5 CONCLUSION

Per-IC quantization method offers a simple yet effective resolution to the activation outlier challenge by strategically isolating the sensitive weights in the IC direction. This methodology is further advanced by our Adaptive Dimensions (AdaDim) framework, which showcases adaptability to varying quantization sensitivities. Experimental results underline AdaDim's effectiveness, evinced by notable performance gains in both base and instruction-tuned LLMs across diverse language modeling benchmarks. Through this work, we hope to make a step forward in the practicality and accessibility of LLMs in real-world applications.

## ACKNOWLEDGMENTS

We thank our colleagues from the AI Efficiency team at NAVER Cloud for constructive feedback, especially Gunho Park, the author of LUT-GEMM (Park et al., 2022), for helping with the kernel implementation. We also thank Ji Lin and Elias Frantar for fruitful discussions.

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

# A  FURTHER DISCUSSIONS

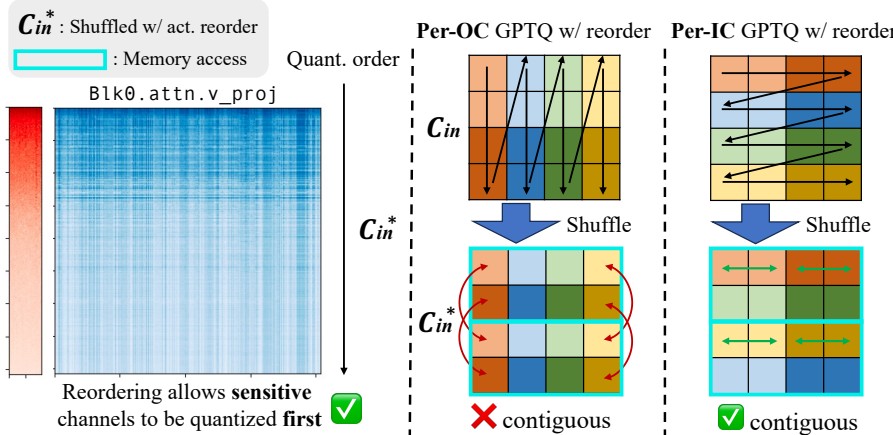

Figure 8: **Activation reordering** allows important (sensitive) channels to be quantized first. This prevents salient channels from being updated to compensate for the quantization error of non-salient ones. However in the standard per-OC scheme, activation reordering causes quantization scales to be not contiguous, causing inefficient memory access. With per-IC, quantization scales are placed in a row-major order, which allows scales to be contiguous even after reordering.

## A.1  ACTIVATION REORDERING FOR FREE

A crucial nature of the GPTQ algorithm is that it prioritizes the weights that are quantized first, since the error of earlier quantized weights is compensated by later quantized weights. Activation reordering trick can help sensitive weights be quantized first, but it is only hardware-efficient with the static groups constraint. As the static groups setting *pre-determines* all quantization parameters *independent* of the weight update step, per-OC GPTQ with hardware-efficient static reordering leads to suboptimal accuracy (Table 2b). In contrast, per-IC GPTQ can decouple quantization parameter's memory contiguity from the usage of activation reordering, thanks to the row-major layout (Figure 8). With per-IC quantization, we can maximize the performance of the GPTQ algorithm by using reordering for free without the static groups constraint.

## A.2  INCOMPATIBILITY WITH AWQ

We found that AdaDim is incompatible to AWQ (Lin et al., 2023), which is another competitive weight-only quantization approach alongside GPTQ (Frantar et al., 2022). The premise of AWQ is that when conventionally grouping the weights within each OC (per-OC quant.), each weight is multiplied by a unique hidden dimension of activations. When activation outliers emerge in a small subset of those channels, the corresponding weights (more sensitive ones) can be identified and scaled up. This effectively transforms the weight distribution such that sensitive weights now lie near the maximum of the distribution, so the min-max quantizer will near-losslessly quantize the scaled up outlier weights. In contrast, per-IC quantization groups weights within each IC, so all the weights end up *sharing the same hidden dimension*; this results in all weights being *scaled up equally*, which essentially nullifies AWQ's activation-based weight transformation. Nonetheless, AWQ can indeed be applied to layers where AdaDim uses per-OC quantization, which may further boost performance.

## A.3   RUNTIME AND COMPUTE

Additional runtime and compute cost incurred from adopting AdaDim is minimal. One point of comparison is AWQ (Lin et al., 2023) which searches over the scale and clip parameters with a grid size of 20 each, totaling 40 forward passes per layer. In contrast, AdaDim requires two forward passes per layer, thanks to the small search space of the dimension parameter.

## B   IMPLEMENTATION

```python
def rtn_ada(x:torch.tensor, layer:torch.nn, w_bit:int, group_size:int
                                            ) -> str:
    # Inputs: calib. data x, layer to be quantized, and quant. parameters
    org_out = layer(x)
    org_sd = {k: v.cpu() for k, v in layer.state_dict().items()}
    best_err = float('inf'); best_dim = None; quant_sd = {}
    for dim in ['oc', 'ic']:
        # quantize the weight matrix in place and save it temporarily
        quant_sd[dim] = per_channel_quantize(weight, dim, w_bit,
                                             group_size)
        quant_out = layer(x)
        recon_err = (org_out - quant_out).float().pow(2).mean().item()
        if recon_err < best_err:
            best_dim = dim
        layer.load_state_dict(org_sd) # recover full precision weight
    layer.load_state_dict(quant_sd[best_dim])
    del quant_sd
    return best_dim

def gptq_ada(x:torch.tensor, layer:torch.nn, w_bit:int, group_size:int):
    weight = layer.weight.data.clone()
    best_dim = rtn_ada(x, layer, w_bit, group_size)
    # run gptq on the searched dimension
    layer.weight.data = run_gptq(weight, best_dim, w_bit, group_size)
```

## C   ADDITIONAL RESULTS

**Ablating GPTQ-ada.** As in Table 5, GPTQ-ada shows the best performance when using 256 samples.

Table 5: Calibration set size ablation on w3g128. We fix the max sequence length to be 512 per sample, thus # of calibration tokens $\sim$ (# of samples $\times$ 512). We use 256 samples for our experiments.

| # of samples | LLaMA-V2-7B | | LLaMA-V2-13B | |
|---|---|---|---|---|
| | MMLU Avg. | CSR Avg. | MMLU Avg. | CSR Avg. |
| 32 | 35.15 | 64.16 | 47.56 | 54.09 |
| 64 | 39.63 | 65.48 | 50.46 | 66.65 |
| 128 | 39.15 | 65.42 | 51.96 | 67.94 |
| 256 | **41.32** | 66.41 | **52.27** | 68.73 |
| 512 | 39.55 | **66.51** | 51.76 | **68.80** |

**INT4** quantization (at least when used with the widely suggested group size of 128 (Dettmers & Zettlemoyer, 2022)) does not have a "winning methodology" that consistently outperforms others (Table 6 & Table 7).

**Additional experiments with SpQR.** To benchmark AdaDim with more modern quantization schemes, we have conducted additional experiments with SpQR (Dettmers et al., 2023), a novel weight-only quantization method featuring small group-wise double quantization and an FP16 outlier sparse representation. Table 8 illustrates that Round-To-Nearest with AdaDim (RTN-ada) achieves comparable performance to SpQR. This similarity arises partly because RTN-ada has a higher average bit-precision than SpQR. For a more balanced comparison regarding average bit-precision with SpQR,

Table 6: Vicuna results across all model scales.

| w4g128 | Vicuna-V1.5-7B | | Vicuna-V1.5-13B | | Vicuna-V1.3-33B | |
| --- | --- | --- | --- | --- | --- | --- |
| | MMLU Avg. | CSR Avg. | MMLU Avg. | CSR Avg. | MMLU Avg. | CSR Avg. |
| FP16 | 50.27 | 67.45 | 56.02 | 69.81 | 59.22 | 71.12 |
| AWQ | 23.06 | 38.39 | **55.68** | 69.36 | 29.73 | 62.49 |
| RTN | 49.18 | 67.36 | 54.88 | 69.00 | 58.32 | **70.82** |
| RTN-ada | **49.57** | **67.55** | 55.17 | 69.40 | 59.04 | 70.63 |
| GPTQ | 49.63 | 66.84 | 55.27 | 69.37 | 58.82 | 70.58 |
| GPTQ-ada | 49.56 | 66.94 | 55.07 | **69.59** | **59.06** | 70.61 |

Table 7: LLaMA results across all model scales.

| w4g128 | LLaMA-V2-7B | | LLaMA-V2-13B | | LLaMA-V1-33B | | LLaMA-V2-70B | |
| --- | --- | --- | --- | --- | --- | --- | --- | --- |
| | MMLU Avg. | CSR Avg. | MMLU Avg. | CSR Avg. | MMLU Avg. | CSR Avg. | MMLU Avg. | CSR Avg. |
| FP16 | 45.98 | 67.93 | 55.61 | 70.33 | 58.46 | 72.96 | 69.29 | 74.96 |
| AWQ | **45.62** | 67.78 | **54.59** | 69.77 | 57.86 | 72.85 | 68.72 | 74.68 |
| RTN | 44.54 | 67.84 | 54.43 | 69.82 | 57.59 | **73.16** | 68.34 | 74.34 |
| RTN-ada | 45.04 | 67.65 | 54.57 | 70.04 | 57.63 | 72.75 | 68.97 | **74.99** |
| GPTQ | 44.93 | **68.18** | 54.20 | 70.00 | **58.00** | 72.51 | **68.99** | 74.66 |
| GPTQ-ada | 45.26 | 67.69 | **54.59** | **70.10** | 57.86 | 72.82 | 68.75 | 74.60 |

we conduct further experiments at lower bit-precisions. From Table 9, it is evident that SpQR outperforms RTN-ada within a similar range of average bit-precision. However, it is crucial to note that SpQR requires an FP16 representation for outliers, necessitating an additional sparse inference kernel, which is not a requirement for AdaDim. We emphasize that by shifting the quantization channel dimensions from output-channel to input-channel, AdaDim can effectively enhance the performance of existing weight-only group-wise quantization methods.

Table 8: Perplexity (PPL) comparison between SpQR (Dettmers et al., 2023) and Round-To-Nearest with AdaDim (RTN-ada) on the Wikitext2 and C4 datasets, employing a 4-bit channel-wise configuration with grouping size of 128. The lower PPL, the better.

| Method | Group Size | Avg Bits | LLaMA-V1-7B Wikitext2 / C4 | LLaMA-V1-33B Wikitext2 / C4 | LLaMA-V1-65B Wikitext2 / C4 |
| --- | --- | --- | --- | --- | --- |
| FP16 | - | 16 | 5.68 / 7.08 | 4.10 / 5.98 | 3.53 / 5.62 |
| RTN-ada | 128 | 4.125 | **5.80 / 7.22** | **4.19 / 6.05** | **3.61 / 5.68** |
| SpQR | 128 | 3.9 | 5.87 / 7.28 | 4.25 / 6.08 | 3.68 / 5.70 |

Table 9: Perplexity (PPL) comparison between SpQR (Dettmers et al., 2023) and Round-To-Nearest with AdaDim (RTN-ada) on the Wikitext2 and C4 datasets, employing a 3-bit channel-wise configuration with various grouping sizes. The lower PPL, the better.

| LLaMA-V1-65B | Group Size | Avg Bits | Wikitext2 [PPL] | C4 [PPL] |
| --- | --- | --- | --- | --- |
| FP16 | - | 16 | 3.53 | 5.62 |
| RTN-ada | 16 | 4.00 | 3.74 | 5.75 |
| RTN-ada | 32 | 3.50 | 3.82 | 5.80 |
| RTN-ada | 64 | 3.25 | 3.91 | 5.87 |
| SpQR | 16 | 3.63 | 3.74 | 5.73 |

# D  PER-IC KERNEL

Table 10 compares the latency of our per-IC kernel against the cuBLAS baseline and other per-OC kernels like OPTQ (Frantar et al., 2022), AWQ (Lin et al., 2023), and LUT-GEMM (Park et al., 2022). Although it shows slower latency compared to per-OC kernels due to its sub-optimal status, it's noteworthy that our per-IC approach is selectively applied to layer-wise manner and enhances the accuracy performance of per-OC baselines, as detailed in the paper.

Table 10: Latency comparison of the first FFN layer on OPT-175B model with 3-bit and 4-bit precision and corresponding kernel selections with hidden dimension size (matrix size, $m$) is set to 12288 and various group sizes on A100-80GB-GPU.

| Method | Group Size | Weight ($4m \times m$) | Input ($m \times 1$) | Output ($4m \times 1$) | Latency [$ms$] |
|---|---|---|---|---|---|
| cuBLAS | | INT4 | FP16 | FP16 | 0.7258 |
| OPTQ | 128 | INT3 | FP16 | FP16 | 0.3599 |
| AWQ | 128 | INT4 | FP16 | FP16 | 0.3238 |
| LUT-GEMM | 128 | INT4 | FP16 | FP16 | 0.2688 |
| LUT-GEMM | 128 | INT3 | FP16 | FP16 | 0.225 |
| **per-IC kernel** | 128 | INT4 | FP16 | FP16 | 0.50477 |
| **(Ours)** | 256 | INT4 | FP16 | FP16 | 0.32926 |
| | 512 | INT4 | FP16 | FP16 | 0.24570 |
| | 2048 | INT4 | FP16 | FP16 | 0.20425 |
| **per-IC kernel** | 128 | INT3 | FP16 | FP16 | 0.47336 |
| **(Ours)** | 256 | INT3 | FP16 | FP16 | 0.29957 |
| | 512 | INT3 | FP16 | FP16 | 0.21457 |
| | 2048 | INT3 | FP16 | FP16 | 0.16735 |

## E  VISUALIZATIONS

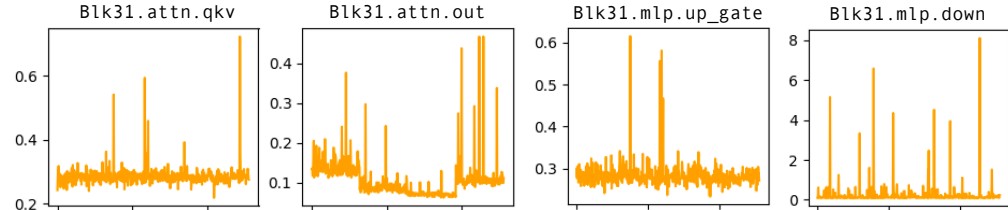

Figure 9: **Activation magnitude** across different modules of LLaMA-V2-7B Block 31. We observe the largest activations before the QKV projection and DOWN projection, which we then selectively apply per-IC quantization to heuristically validate the outlier isolation effect.

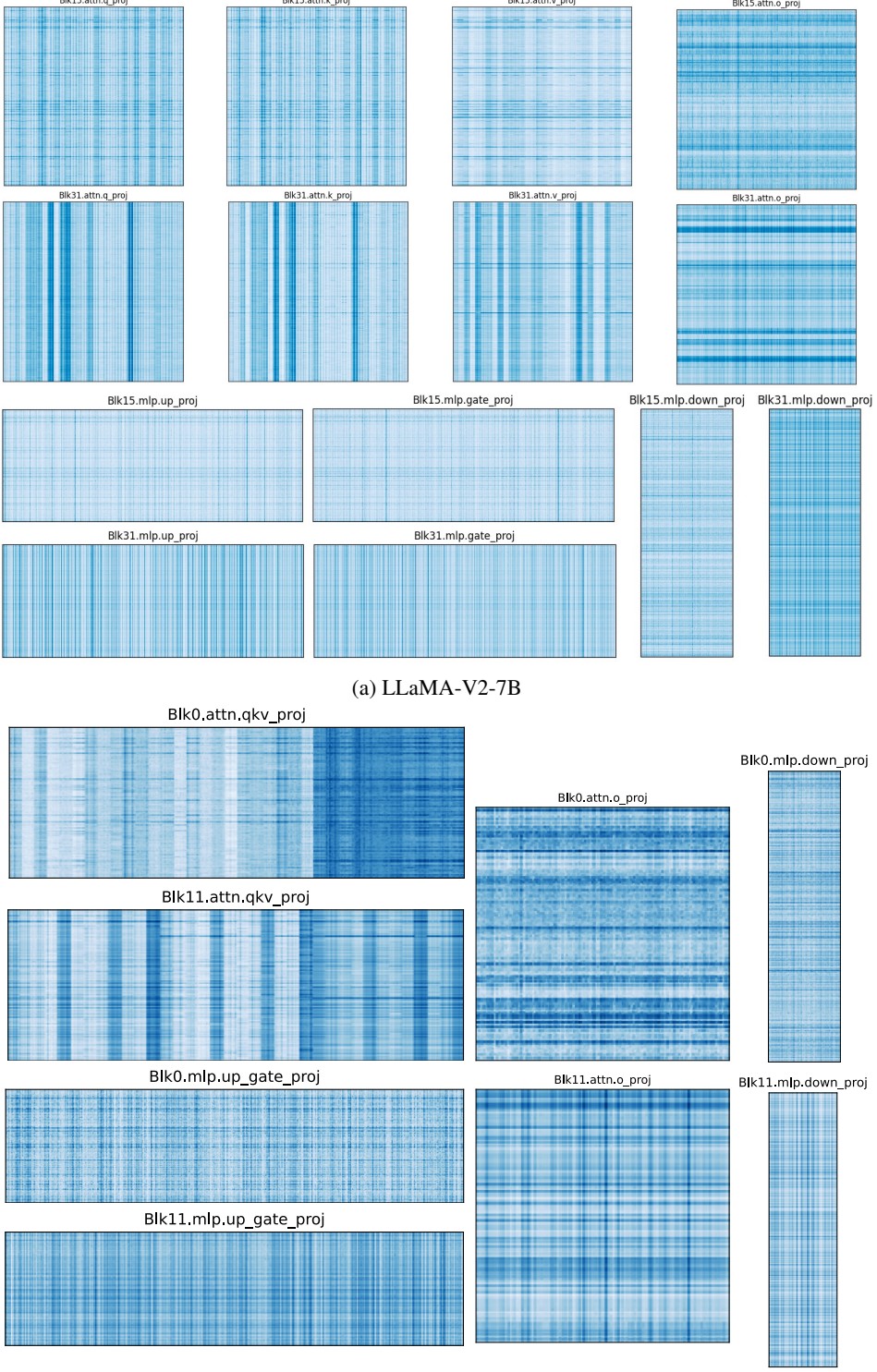

(a) LLaMA-V2-7B

(b) GPT-2. QKV projections are horizontally concatenated.

Figure 10: **Weight sensitivity patterns shown in $(\mathbf{C_{in}}, \mathbf{C_{out}})$ shape.** Both sensitive rows and columns exist across different modules and network depth. This hints at the potential effectiveness of AdaDim's versatile quantization scheme to other backbones such as GPT-2 (Radford et al., 2019).

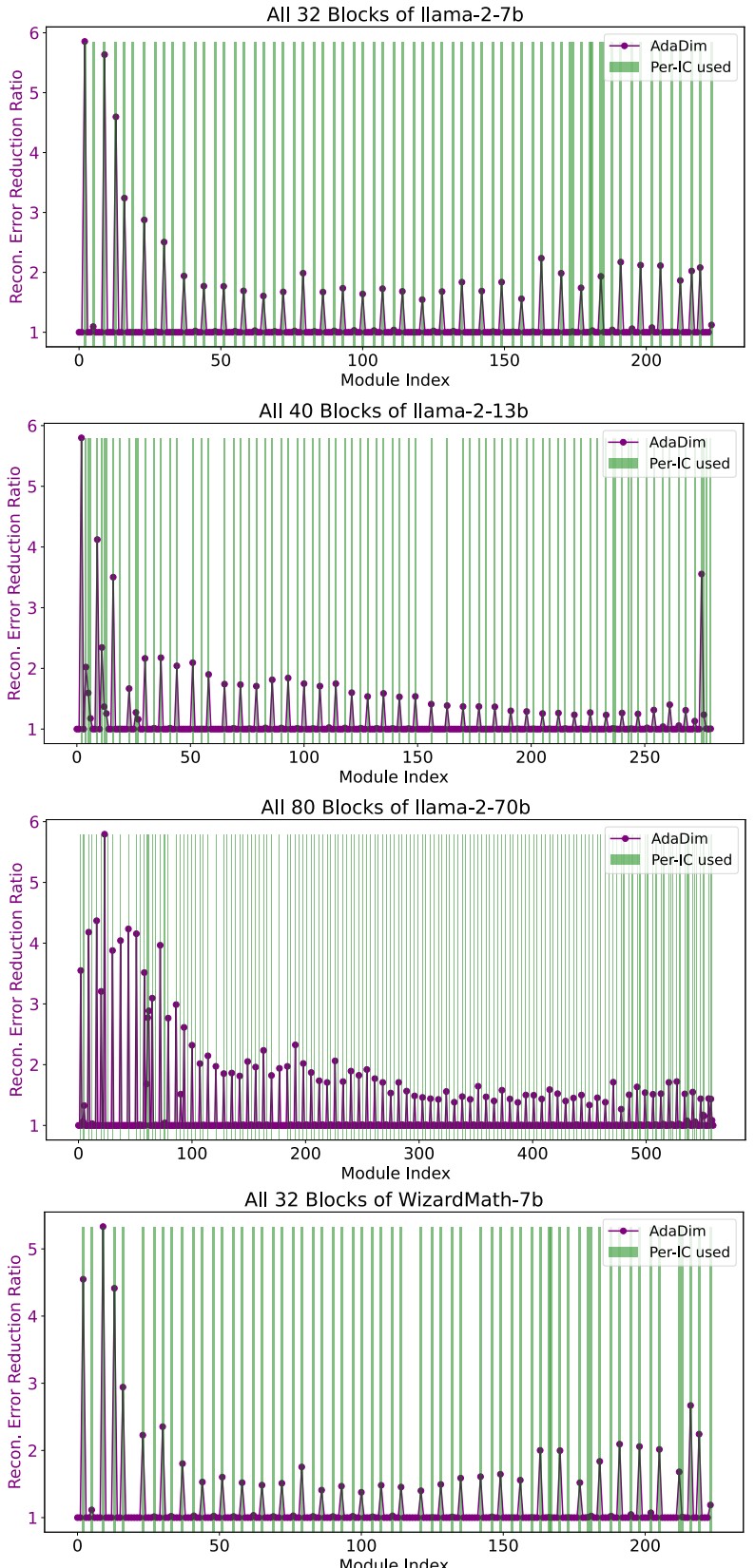

Figure 11: Full visualization of adaptively selected per-IC quantization decisions and the subsequent reconstruction error savings for INT3 with group size 128.

