# OpenReview forum: "Rethinking Channel Dimensions to Isolate Outliers for Low-bit Weight Quantization of Large Language Models"
_ICLR.cc/2024/Conference — ICLR 2024 poster_

### Official Review · Reviewer_horP · 2023-10-27

**Soundness:** 3 good
**Presentation:** 1 poor
**Contribution:** 2 fair
**Rating:** 6
**Confidence:** 4

**Summary:**

This paper proposes a new quantization scheme for LLMs. Particularly, they propose to choose quantizers based on input channels rather than output channels, as it is usually done. They show that this can to some degree reduce the effect of weight outliers on the quality of the quantized network.

**Strengths:**

The numerical results suggest that this quantization framework can help reduce the loss of accuracy from quantization.

**Weaknesses:**

- First, I think the presentation of the paper is very hard to follow. The authors choose to write a paragraph rather using mathematical notations that can make the paper easier to follow. At core, given the activation as $WX$ with $W\in R^{q\times p}$, they propose to have $p$ quantizers where quantizer $i$ corresponds to column $i$ of $W$, rather than having $q$ of them each corresponding to a row of $W$. Other examples include using the term "Activation". Is activation referring to $WX$ or $X$? Or $W$ which are the weights? Isn't this work trying to circumvent outliers in $W$? Then why they talk about activation outliers in page 3? The method does not even seem to detect outliers, then why talk so much about outliers?  Overall, I find the paper confusing and hard to follow.

- It seems transposing the direction of quantization helps with accuracy. But then what happens in terms of implementing this method in practice? Note that this per-IC quantization method is not a drop-in replacement for per-OC, or at least if it is, that requires to be substantiated with experiments. As far as I could tell there are no experiments on hardware implementation and inference time performance of the proposal. The hardware implementation is very briefly mentioned in Section 3.2, but the explanation actually creates more questions than answers, as it contain a bunch of new notations that are never properly defined. The authors say "With per-IC quantization, we can maximize the performance of the GPTQ algorithm by using reordering for free without the static groups constraint." which again I don't think is numerically supported.

Overall, I think the paper needs some rewriting to make it easier to understand, and several experiments to show the benefits of per-IC in inference time.

**Questions:**

See above.

---

> ### Author Response · Authors · 2023-11-20
>
> Dear Reviewer F2mg,
> We appreciate your constructive comments.
>
> --------------
>
> ### **[Weakness 1]**
> We acknowledge any confusion caused by the initial version of our manuscript. We are thankful to the reviewer 'horP' for identifying key revision areas. These, along with the feedback from reviewers 'JAtq' and 'ubBB', will be addressed in the revised manuscript. We appreciate the identification of areas for enhancement in our presentation.
>
> Responding to the reviewer’s query regarding 'activation outliers', we reference the memory-wall problem [1] associated with large language models, where memory, rather than computation, is the primary bottleneck. For this reason, weight-only quantization methods employ low-bit quantization with grouping techniques [2 - 4]. Employing smaller group sizes enhances the isolation of outliers in large language models [2]. In this regard, our approach suggests that the direction of quantization grouping significantly affects the accuracy of quantized large language models, a claim supported by analyzing the relationship between weight sensitivity and activation outliers.
>
> For a comprehensive explanation, we structurally analyze this relationship, paralleling the methodology of [1] as mentioned in our initial paper. Following [1], we use a small calibration data set to assess weight sensitivity, which is particularly reactive to specific activation outliers. We align with the perspectives of the SpQR paper [2] in this regard. Given the variation in weight sensitivity patterns across the network's depth, as shown in Figure 2 of our paper, AdaDim employs the reconstruction error metric to assess the most effective dimension (per-OC or per-IC) on a layer-by-layer basis. By incorporating the input dimension as a new design parameter, AdaDim achieves non-negligible performance improvements over baseline methods without AdaDim. To provide a clearer understanding of our approach, particularly concerning activation outliers, we will make clearer revisions to the manuscript. Thank you for your valuable feedback again.
>
> ----------------------

---

> ### Author Response · Authors · 2023-11-20
>
> ### **[Weakness 2 - a] It seems transposing the direction of quantization helps with accuracy. But then what happens in terms of implementing this method in practice? Note that this per-IC quantization method is not a drop-in replacement for per-OC, or at least if it is, that requires to be substantiated with experiments. As far as I could tell there are no experiments on hardware implementation and inference time performance of the proposal. The hardware implementation is very briefly mentioned in Section 3.2, but the explanation actually creates more questions than answers, as it contain a bunch of new notations that are never properly defined.**
>
> We appreciate your comments that have contributed to the qualitative enhancement of our paper. In this paper, we primarily focus on the algorithmic aspects of weight-only quantization for large language models, with a particular emphasis on isolating outliers. However, we acknowledge the reviewer’s concerns regarding the computational overhead of per-IC quantization proposed in our work. To address this issue, we have naively implemented a per-IC (Input Channel) kernel based on the LUT-GEMM [3, 6] kernel. We utilize the implementation of LUT-GEMM, Look up table-based (LUT-based) matrix multiplication, as described in [3]. LUT-based matrix multiplication involves precomputing possible multiplication outcomes between quantized weights and activations, storing them in a lookup table. During computation, the corresponding operation results are efficiently retrieved from the lookup table using the weight values as keys, enabling efficient GEMM operations.
>
> For per-IC quantization, a similar approach can be applied. During LUT generation, each per-IC quantization scaling factor can be multiplied in advance with activations to create the lookup table. In cases of group quantization, adjusting tile sizes to fit the groups and generating LUTs on a tile-by-tile basis allows for a similar application of this method. Unfortunately, due to time and resource limitations, we couldn't perform experiments with a fully optimized kernel to evaluate the optimal latency of our approach. Nevertheless, Table A compares the latency of our per-IC kernel against the cuBLAS baseline and other per-OC kernels like OPTQ [4], AWQ [7], and LUT-GEMM [3]. As demonstrated in Table A, our per-IC kernel exhibits faster latency than the cuBLAS baseline. Although it shows slower latency compared to per-OC kernels due to its sub-optimal status, it's noteworthy that our per-IC approach is selectively applied to layer-wise manner and enhances the accuracy performance of per-OC baselines, as detailed in the paper. This indicates that our method, which shifts channel dimensions from output channel to input channel, not only achieves accuracy improvement but also accelerates inference latency compared to cuBLAS baselines. Further investigation into a well-optimized per-IC kernel is a critical direction for our future research. We will include detailed information on kernel implementation and experimental results in the revised manuscript. We are grateful for the insightful feedback provided. We will also provide implemented kernel code for supplementary.
>
>
> Table A. Latency comparison of the first FFN layer on OPT-175B model with 3-bit and 4-bit precision and corresponding kernel selections with hidden dimension size (matrix size, m) is set to 12288 and various group sizes on A100-80GB-GPU.
> | method               | group size    | Weight (4m x m) | Input (m x 1) | Output (4m x 1) | Latency [ms] |
> | -------------------- | ---- | ------ | ----- | ------ | ------------ |
> | cuBLAS               |      | FP16   | FP16  | FP16   | 0.7258       |
> | OPTQ                 | 128  | INT3   | FP16  | FP16   | 0.3599       |
> | AWQ                  | 128  | INT4   | FP16  | FP16   | 0.3238       |
> | LUT-GEMM             | 128  | INT4   | FP16  | FP16   | 0.2688       |
> | LUT-GEMM             | 128  | INT3   | FP16  | FP16   | 0.225        |
> |                      |      |        |       |        |              |
> | per-IC kernel (Ours) | 128  | INT4   | FP16  | FP16   | 0.57701      |
> | per-IC kernel (Ours) | 256  | INT4   | FP16  | FP16   | 0.39219      |
> | per-IC kernel (Ours) | 512  | INT4   | FP16  | FP16   | 0.30298      |
> | per-IC kernel (Ours) | 2048 | INT4   | FP16  | FP16   | 0.27206      |
> |                      |      |        |       |        |              |
> | per-IC kernel (Ours) | 128  | INT3   | FP16  | FP16   | 0.53928      |
> | per-IC kernel (Ours) | 256  | INT3   | FP16  | FP16   | 0.35392      |
> | per-IC kernel (Ours) | 512  | INT3   | FP16  | FP16   | 0.26456      |
> | per-IC kernel (Ours) | 2048 | INT3   | FP16  | FP16   | 0.20333      |

---

> ### Author Response · Authors · 2023-11-20
>
> ### **[Weakness 2 - b] The authors say "With per-IC quantization, we can maximize the performance of the GPTQ algorithm by using reordering for free without the static groups constraint." which again I don't think is numerically supported.**
>
> GPTQ features two options: reordering and static-groups. Conventionally, GPTQ quantizes weights sequentially from the first to the last column, a process that does not account for activations, resulting in sub-optimal performance [10]. To address this, reordering was proposed, which reorganizes the quantization process to start from columns with larger activations down to those with smaller ones, thereby enhancing accuracy. However, reordering introduces non-contiguous quantization scaling factors, leading to irregular DRAM access and additional overhead [1, 9], as evidenced in our paper's Figure 3 (center), Section 2.3 of AWQ-v1-arxiv [9], and the footnote of Table 1 in SqueezeLLM [1].
>
> To mitigate the overhead from reordering, the static-groups method was introduced. This method, applied after reordering, aligns the quantization scale contiguously, effectively removing the extra overhead. Nonetheless, this approach diminishes the accuracy gains achieved through reordering, as demonstrated in Table 2(b) of our paper.
>
> For per-IC quantization, the situation is distinct. Post-reordering, the quantization scaling factors remain contiguous even without applying static-groups, as shown on the right side of Figure 3 in our paper. Therefore, while per-OC quantization may initially improve accuracy with reordering, this gain is subsequently reduced by the application of static-groups. In contrast, per-IC quantization, maintaining hardware efficiency solely through reordering, offers the benefit of accuracy improvement, setting it apart from per-OC quantization.
>
> ----------
>
> ### **Reference**
>
> [1] Kim, Sehoon, et al. "SqueezeLLM: Dense-and-Sparse Quantization." arXiv preprint arXiv:2306.07629 (2023).
> [2] Dettmers, Tim, et al. "SpQR: A Sparse-Quantized Representation for Near-Lossless LLM
> Weight Compression." arXiv preprint arXiv:2306.03078 (2023).
> [3] Park, Gunho, et al. "nuQmm: Quantized matmul for efficient inference of large-scale generative language models." arXiv preprint arXiv:2206.09557 (2022).
> [4] Frantar, Elias, et al. "OPTQ: Accurate quantization for generative pre-trained transformers." The Eleventh International Conference on Learning Representations. 2022.
> [5] https://github.com/NVIDIA/FasterTransformer.
> [6] Jeon, Yongkweon, et al. "Biqgemm: matrix multiplication with lookup table for binary-coding-based quantized dnns." SC20: International Conference for High Performance Computing, Networking, Storage and Analysis. IEEE, 2020.
> [7] Lin, Ji, et al. "AWQ: Activation-aware Weight Quantization for LLM Compression and Acceleration." arXiv preprint arXiv:2306.00978 (2023).
> [8] https://developer.nvidia.com/cublas
> [9] https://arxiv.org/pdf/2306.00978v1.pdf
> [10] https://github.com/IST-DASLab/gptq

---

> ### Author Response · Authors · 2023-11-21
>
> Dear Reviewer,
>
> We sincerely appreciate your valuable feedback and remain open to further discussions.
>
> In addition to our previous responses, we would like to present further comparative results of our approach:
>
> To enhance our analysis with more baseline methods, we have conducted additional experiments with SpQR [2], a novel weight-only quantization method featuring small group-wise double quantization and an FP16 outlier sparse representation. Table B illustrates that Round-To-Nearest with ada (RTN-ada) achieves performance comparable to SpQR. This similarity arises partly because RTN-ada adopts a more average bit-precision than SpQR in these tests. For a more balanced comparison regarding average bit-precision with SpQR, we undertook further experiments at lower bit-precisions, with the results presented in Table C. From Table C, it is evident that SpQR outperforms RTN-ada within a similar range of average bit-precision. However, it is crucial to note that SpQR requires an FP16 representation for outliers, necessitating an additional sparse inference kernel, which is not a requirement for AdaDim. We emphasize that by shifting the quantization channel dimensions from output-channel to input-channel, AdaDim can effectively enhance the performance of existing weight-only group-wise quantization methods.
>
> Should you have any questions or need further clarifications regarding our rebuttal, please feel free to reach out. We are committed to addressing any concerns and providing further explanations to elucidate potential ambiguities in our findings.
>
>
> Table B. Perplexity (PPL) comparison between SpQR [2] and Round-To-Nearest with AdaDim (RTN-ada) on the Wikitext2 and C4 datasets, employing a 4-bit channel-wise configuration with grouping size of 128. The lower PPL, the better.
> |         |           |     LLaMA-V1-7B     |     LLaMA-V1-33B    |     LLaMA-V1-65B    |
> |:-------:|:---------:|:-------------------:|:-------------------:|:-------------------:|
> |  Method | Avg bits. | Wikitext2 [PPL] / C4 [PPL] | Wikitext2 [PPL] / C4 [PPL] | Wikitext2 [PPL] / C4 [PPL] |
> |   FP16  |     16    |     5.68 / 7.08     |     4.10 / 5.98     |     3.53 . 5.62     |
> | RTN-ada |   4.125   | **5.80** / **7.22** | **4.19** / **6.05** | **3.61** / **5.68** |
> |   SpQR  |    3.9    |     5.87 / 7.28     |     4.25 / 6.08     |     3.68 / 5.70     |
>
> Table C. Perplexity (PPL) comparison between SpQR [2] and Round-To-Nearest with AdaDim (RTN-ada) on the Wikitext2 and C4 datasets, employing a 3-bit channel-wise configuration with various grouping sizes. The lower PPL, the better.
> | LLaMA-V1-65B | Group-size | Avg Bits | Wikitext2 [PPL] | C4 [PPL] |
> |:------------:|:----------:|:--------:|:----------:|:------:|
> |     FP16     |      -     |    16    |    3.53    |  5.62  |
> |    RTN-ada   |     16     |   4.00   |    3.74    |  5.75  |
> |    RTN-ada   |     32     |   3.50   |    3.82    |  5.80  |
> |    RTN-ada   |     64     |   3.25   |    3.91    |  5.87  |
> |     SpQR     |     16     |   3.63   |    3.74    |  5.73  |

---

> > ### Comment · Reviewer_horP · 2023-11-22
> >
> > Thank you for detailed response. I have increased my evaluation.

---

> ### Author Response · Authors · 2023-11-23
>
> We are pleased to have addressed your concerns and questions. We will ensure that your comments are incorporated into the revised manuscript and appreciate the guidance you've provided for our future research endeavors.
>
> We kindly note that our per-IC kernel is further optimized during the rebuttal process.
> Below is the updated Table of kernel latency.
>
> Table A. Latency comparison of the first FFN layer on OPT-175B model with 3-bit and 4-bit precision and corresponding kernel selections with hidden dimension size (matrix size, m) is set to 12288 and various group sizes on A100-80GB-GPU.
> | method               | group size    | Weight (4m x m) | Input (m x 1) | Output (4m x 1) | Latency [ms] |
> | -------------------- | ---- | ------ | ----- | ------ | ------------ |
> | cuBLAS               |      | FP16   | FP16  | FP16   | 0.7258       |
> | OPTQ                 | 128  | INT3   | FP16  | FP16   | 0.3599       |
> | AWQ                  | 128  | INT4   | FP16  | FP16   | 0.3238       |
> | LUT-GEMM             | 128  | INT4   | FP16  | FP16   | 0.2688       |
> | LUT-GEMM             | 128  | INT3   | FP16  | FP16   | 0.225        |
> |                      |      |        |       |        |              |
> | per-IC kernel (Ours) | 128  | INT4   | FP16  | FP16   | 0.50477      |
> | per-IC kernel (Ours) | 256  | INT4   | FP16  | FP16   | 0.32926      |
> | per-IC kernel (Ours) | 512  | INT4   | FP16  | FP16   | 0.24570      |
> | per-IC kernel (Ours) | 2048 | INT4   | FP16  | FP16   | 0.20425      |
> |                      |      |        |       |        |              |
> | per-IC kernel (Ours) | 128  | INT3   | FP16  | FP16   | 0.47336      |
> | per-IC kernel (Ours) | 256  | INT3   | FP16  | FP16   | 0.29957      |
> | per-IC kernel (Ours) | 512  | INT3   | FP16  | FP16   | 0.21457      |
> | per-IC kernel (Ours) | 2048 | INT3   | FP16  | FP16   | 0.16735      |

---

### Official Review · Reviewer_JAtq · 2023-10-30

**Soundness:** 2 fair
**Presentation:** 3 good
**Contribution:** 2 fair
**Rating:** 6
**Confidence:** 4

**Summary:**

This work introduces input-channel quantization and adaptive dimension (AdaDim) for input or output channel selection. The authors analyzes the outlier distributions and observed the traditional output-channel quantization is sub-optimal on some layers. Neither input nor output channel quantization is universally optimal, so the authors proposed AdaDim for channel selection. The performance improvement over SOTA 3-bit quantization is noticeable, but the improvement over 4-bit quantization is very minor.

**Strengths:**

1. The analysis on input/output channel sensitivity is very interesting (figure 2 and 9).

2. Performance improvement on WizMath and WizCoder is pretty impressive (Table 4).

3. While input channel quantization isn't new (weakness 1), AdaDim is novel.

**Weaknesses:**

1. Exploring non-output channel quantization isn't new. See [1] that explores input-channel and arbitrary combination of row and column groups for quantization.

2. All evaluations in this paper use sub-channel quantization with group size of 128. While this is the SOTA configuration, I suggest authors to compare per-channel quantization (without groups) which might better demonstrate the advantage of input channel quantization over output channel quantization for certain layers. This is because the analysis was compare channel-wise sensitivity instead of sub-channel sensitivity.

[1] Yuan, Z., Chen, Y., Xue, C., Zhang, C., Wang, Q. and Sun, G., 2021. Ptq-sl: Exploring the sub-layerwise post-training quantization. arXiv preprint arXiv:2110.07809.

**Questions:**

1. Why did the sensitivity is done in channel-wise, but the experiments were done with sub-channel?

2. If you perform the sensitivity analysis based on the sub-channels, do you still see a large gap between input/output channel quantization?

---

> ### Author Response · Authors · 2023-11-20
>
> Dear reviewer JAtq,
> We are grateful for the valuable feedback provided by the reviewer. In this response, we thoroughly address each of your comments and suggestions.
>
> ---------------
>
> ### **[Weakness 1] Exploring non-output channel quantization isn't new. See [1] that explores input-channel and arbitrary combination of row and column groups for quantization.**
>
> In response to the reviewer's comment, we acknowledge that the non-output channel quantization concept itself is not a new concept. However, it is noteworthy that our proposed approach, which involves shifting the quantization channel dimension from output-channel to input-channel specifically for isolating outliers in large language models, has not yet been explored. We appreciate the reference and will include this citation in the revised manuscript. Thank you for bringing this to our attention.
>
> ---------------
> ### **[Weakness 2] All evaluations in this paper use sub-channel quantization with group size of 128. While this is the SOTA configuration, I suggest authors to compare per-channel quantization (without groups) which might better demonstrate the advantage of input channel quantization over output channel quantization for certain layers. This is because the analysis was compare channel-wise sensitivity instead of sub-channel sensitivity.**
>
> In light of the memory-wall problem [1] associated with large language models, where memory is the primary bottleneck rather than computation, weight-only quantization methods employ low-bit quantization with grouping techniques. Using smaller group sizes allows these techniques to more effectively isolate outliers in large language models [2]. Our approach posits that the direction of quantization grouping influences the accuracy of quantized large language models. Nonetheless, responding to the reviewer's suggestion and to provide clearer insights, we have conducted additional experiments, as presented in Table A and B. These experiments compare per-channel quantization (without groups) using AdaDim on LLaMAs. As indicated in Table A and B, regardless of whether RTN is with or without AdaDim, 3-bit weight-only quantization without grouping results in non-negligible performance degradation. However, for 4-bit quantization, RTN with AdaDim exhibits marginally better performance than RTN without AdaDim in terms of perplexity. Additionally, based on the findings in Table A and B, AdaDim appears less effective in isolating outliers without grouping techniques. This experiment confirms the importance of the grouping technique in isolating outliers in LLMs which are the same perspective of SpQR [2]. We appreciate the feedback for enhancing the clarity of our presentation and will address this point in the revised manuscript.
>
> Table A. Perplexity (PPL) comparison between RTN and RTN-ada on the Wikitext2 and C4 datasets, employing a 4-bit channel-wise configuration without grouping scales. The lower PPL, the better.
> |         | LLaMA-V1-7B         | LLaMA-V1-33B        |
> |---------|---------------------|---------------------|
> | Method  | Wikitext2 [PPL]  / C4 [PPL] | Wikitext2 [PPL] / C4 [PPL] |
> | FP16    | 5.68 / 7.08         | 4.10 / 5.98         |
> | RTN     | 6.29 / 7.73         | 4.54 / 6.33         |
> | RTN-ada | 6.22 / 7.66 | 4.50 / 6.30 |
>
> Table B. Average score comparison between RTN and RTN-ada on the MMLU and Common sense reasoning (CSR) benchmark, employing a 3-bit channel-wise configuration without grouping scales.
> |         |  LLaMA-V2-7B  |  LLaMA-V2-13B |  LLaMA-V1-33B |  LLaMA-V2-70B |
> |:-------:|:-------------:|:-------------:|:-------------:|:-------------:|
> |  Method |   MMLU(%) / CSR(%)  |   MMLU(%) / CSR(%)  |   MMLU(%) / CSR(%)  |   MMLU(%) / CSR(%)  |
> |   FP16  | 45.98 / 67.93 | 55.61 / 70.33 | 58.46 / 72.96 |   69.29 / ?   |
> |   RTN   | 24.57 / 42.62 | 27.11 / 55.82 | 27.13 / 48.12 | 34.56 / 59.72 |
> | RTN-ada | 27.42 / 53.83 | 25.63 / 46.19 | 26.56 / 51.94 | 47.71 / 64.53 |
>
> ---------------
>
> ### **[Question 1] Why did the sensitivity is done in channel-wise, but the experiments were done with sub-channel?**
>
> Our sensitivity analysis reveals that the "sensitivity dimension (Input Channel or Output Channel) can vary across the network's depth." This is the rationale behind AdaDim's use of the reconstruction error metric to determine which dimension is more effective on a layer-by-layer basis, regardless whether group-wise quantization is applied or not. We acknowledge that this point may have been unclear in our initial manuscript and will make the necessary revisions to clarify it. We appreciate your comments that have contributed to the qualitative enhancement of our paper.
>
> ---------------
> ### **Reference**
> [1] Kim, Sehoon, et al. "SqueezeLLM: Dense-and-Sparse Quantization." arXiv preprint arXiv:2306.07629 (2023).
> [2] Dettmers, Tim, et al. "SpQR: A Sparse-Quantized Representation for Near-Lossless LLM Weight Compression." arXiv preprint arXiv:2306.03078 (2023).

---

> ### Author Response · Authors · 2023-11-20
>
> ### **[Question 2] If you perform the sensitivity analysis based on the sub-channels, do you still see a large gap between input/output channel quantization?**
>
> As previously discussed in Weakness 2, the scale of sub-channels plays a critical role in the performance of quantized large language models (LLMs). In scenarios that utilize these sub-channels, AdaDim demonstrates a non-negligible improvement in performance compared to baselines without AdaDim. However, it's important to note that without the grouping technique, the beneficial impact of AdaDim on the accuracy of quantized LLMs becomes much less evident. This underscores the critical role of grouping techniques in boosting the efficacy of AdaDim for optimizing quantization strategies. We acknowledge this insightful feedback and will include these clarifications in the revised manuscript.

---

> ### Author Response · Authors · 2023-11-21
>
> Dear reviewer,
>
> We genuinely value your feedback and are always open to further discussions. Additionally, during the rebuttal process, we would like to highlight two additional implications of our approach:
>
> To validate the feasibility of per-IC quantization kernel, we have naively implemented a per-IC (Input Channel) kernel based on the LUT-GEMM [3, 4] kernel. We utilize the implementation of LUT-GEMM, Look up table-based (LUT-based) matrix multiplication, as described in [3]. LUT-based matrix multiplication involves precomputing possible multiplication outcomes between quantized weights and activations, storing them in a lookup table. During computation, the corresponding operation results are efficiently retrieved from the lookup table using the weight values as keys, enabling efficient GEMM operations.
> For per-IC quantization, a similar approach can be applied. During LUT generation, each per-IC quantization scaling factor can be multiplied in advance with activations to create the lookup table. In cases of group quantization, adjusting tile sizes to fit the groups and generating LUTs on a tile-by-tile basis allows for a similar application of this method. Unfortunately, due to time and resource limitations, we couldn't perform experiments with a fully optimized kernel to evaluate the optimal latency of our approach. Nevertheless, Table E compares the latency of our per-IC kernel against the cuBLAS [7] baseline and other per-OC kernels like OPTQ [5], AWQ [6], and LUT-GEMM [3]. As demonstrated in Table E, our per-IC kernel exhibits faster latency than the cuBLAS baseline. Although it shows slower latency compared to per-OC kernels due to its sub-optimal status, it's noteworthy that our per-IC approach is selectively applied to layer-wise manner and enhances the accuracy performance of per-OC baselines, as detailed in the paper. This indicates that our method, which shifts channel dimensions from output channel to input channel, not only achieves accuracy improvement but also accelerates inference latency compared to cuBLAS baselines. Further investigation into a well-optimized per-IC kernel is a critical direction for our future research. We will include detailed information on kernel implementation and experimental results in the revised manuscript.
>
> Table E. Latency comparison of the first FFN layer on OPT-175B model with 3-bit and 4-bit precision and corresponding kernel selections with hidden dimension size (matrix size, m) is set to 12288 and various group sizes on A100-80GB-GPU.
> | method               | group size    | Weight (4m x m) | Input (m x 1) | Output (4m x 1) | Latency [ms] |
> | -------------------- | ---- | ------ | ----- | ------ | ------------ |
> | cuBLAS               |      | FP16   | FP16  | FP16   | 0.7258       |
> | OPTQ                 | 128  | INT3   | FP16  | FP16   | 0.3599       |
> | AWQ                  | 128  | INT4   | FP16  | FP16   | 0.3238       |
> | LUT-GEMM             | 128  | INT4   | FP16  | FP16   | 0.2688       |
> | LUT-GEMM             | 128  | INT3   | FP16  | FP16   | 0.225        |
> |                      |      |        |       |        |              |
> | per-IC kernel (Ours) | 128  | INT4   | FP16  | FP16   | 0.50477      |
> | per-IC kernel (Ours) | 256  | INT4   | FP16  | FP16   | 0.32926      |
> | per-IC kernel (Ours) | 512  | INT4   | FP16  | FP16   | 0.24570      |
> | per-IC kernel (Ours) | 2048 | INT4   | FP16  | FP16   | 0.20425      |
> |                      |      |        |       |        |              |
> | per-IC kernel (Ours) | 128  | INT3   | FP16  | FP16   | 0.47336      |
> | per-IC kernel (Ours) | 256  | INT3   | FP16  | FP16   | 0.29957      |
> | per-IC kernel (Ours) | 512  | INT3   | FP16  | FP16   | 0.21457      |
> | per-IC kernel (Ours) | 2048 | INT3   | FP16  | FP16   | 0.16735      |

---

> ### Author Response · Authors · 2023-11-21
>
> To expand our comparison with more baseline methods, we conducted additional experiments with SpQR [8], which are novel weight-only quantization methods incorporating small group-wise double quantization and FP16 outlier sparse representation. Table C shows that Round-To-Nearest with ada (RTN-ada) achieves performance comparable to SpQR. This similarity arises partly because RTN-ada adopts a more average bit-precision than SpQR in these tests. For a more balanced comparison regarding average bit-precision with SpQR, we undertook further experiments at lower bit-precisions, with the results presented in Table D. According to Table D, SpQR surpasses RTN with ada in a similar and lower average bit-precision range. However, it's noteworthy that SpQR requires an FP16 representation for outliers, necessitating an additional sparse inference kernel, which is not a requirement for AdaDim. We would like to emphasize that by shifting the focus of quantization channel dimensions from output-channel to input-channel with AdaDim, the performance of current weight-only group-wise quantization methods can be effectively enhanced.
>
> Table C. Perplexity (PPL) comparison between SpQR [8] and Round-To-Nearest with AdaDim (RTN-ada) on the Wikitext2 and C4 datasets, employing a 4-bit channel-wise configuration with grouping size of 128. The lower PPL, the better.
> |         |           |     LLaMA-V1-7B     |     LLaMA-V1-33B    |     LLaMA-V1-65B    |
> |:-------:|:---------:|:-------------------:|:-------------------:|:-------------------:|
> |  Method | Avg bits. | Wikitext2 [PPL] / C4 [PPL] | Wikitext2 [PPL] / C4 [PPL] | Wikitext2 [PPL] / C4 [PPL] |
> |   FP16  |     16    |     5.68 / 7.08     |     4.10 / 5.98     |     3.53 . 5.62     |
> | RTN-ada |   4.125   | **5.80** / **7.22** | **4.19** / **6.05** | **3.61** / **5.68** |
> |   SpQR  |    3.9    |     5.87 / 7.28     |     4.25 / 6.08     |     3.68 / 5.70     |
>
> Table D. Perplexity (PPL) comparison between SpQR [8] and Round-To-Nearest with AdaDim (RTN-ada) on the Wikitext2 and C4 datasets, employing a 3-bit channel-wise configuration with various grouping sizes. The lower PPL, the better.
> | LLaMA-V1-65B | Group-size | Avg Bits | Wikitext2 [PPL] | C4 [PPL] |
> |:------------:|:----------:|:--------:|:----------:|:------:|
> |     FP16     |      -     |    16    |    3.53    |  5.62  |
> |    RTN-ada   |     16     |   4.00   |    3.74    |  5.75  |
> |    RTN-ada   |     32     |   3.50   |    3.82    |  5.80  |
> |    RTN-ada   |     64     |   3.25   |    3.91    |  5.87  |
> |     SpQR     |     16     |   3.63   |    3.74    |  5.73  |
>
> Should you have any inquiries or require clarifications about our rebuttal, please don't hesitate to reach out.
> We are eager to address any concerns and elucidate potential ambiguities in greater depth.
>
>
> ----------
>
> ### **Reference**
> [3] Park, Gunho, et al. "nuQmm: Quantized matmul for efficient inference of large-scale generative language models." arXiv preprint arXiv:2206.09557 (2022).
> [4] Jeon, Yongkweon, et al. "Biqgemm: matrix multiplication with lookup table for binary-coding-based quantized dnns." SC20: International Conference for High Performance Computing, Networking, Storage and Analysis. IEEE, 2020.
> [5] Frantar, Elias, et al. "OPTQ: Accurate quantization for generative pre-trained transformers." The Eleventh International Conference on Learning Representations. 2022.
> [6] Lin, Ji, et al. "AWQ: Activation-aware Weight Quantization for LLM Compression and Acceleration." arXiv preprint arXiv:2306.00978 (2023).
> [7] https://developer.nvidia.com/cublas
> [8] Dettmers, Tim, et al. "SpQR: A Sparse-Quantized Representation for Near-Lossless LLM Weight Compression." arXiv preprint arXiv:2306.03078 (2023).

---

> ### Author Response · Authors · 2023-11-23
>
> Dear Reviewer JAtq,
>
> As the discussion period approaches its conclusion and we have not yet received your comments, we wish to kindly remind you about our recent response. We are particularly eager to hear your thoughts on the additional results we have shared.
>
> Best regards,
> The Authors.

---

### Official Review · Reviewer_ubBB · 2023-10-31

**Soundness:** 3 good
**Presentation:** 3 good
**Contribution:** 2 fair
**Rating:** 6
**Confidence:** 5

**Summary:**

The paper focuses on the weight-only quantization of Large Language Models (LLMs) and claims the case of less than 4 bits has challenges because of the presence of large-magnitude activation outliers. Based on the observation, the authors propose a method called "per-IC quantization" within AdaDim framework, which creates quantization groups within each input channel (IC) rather than the conventional per-output-channel (per-OC). The experiment results show the proposed superiority.

**Strengths:**

1. The high-level idea is good. The idea of "per-IC quantization" is clear and effective solution.

2. This paper is well-written and organized.

3. I agree the issue that accelerating the memory I/O is important in some scenes, and solving this challenge is very valuable in the industry.

**Weaknesses:**

1. The experiment results with activation quantization is expected, and more compared methods such [1] are necessary.

2. The result of A.3 is important, more detailed analysis with diagram is expected.

[1] Smoothquant: Accurate and efficient post-training quantization for large language models

**Questions:**

1. Typo error: The MMLU Avg. (49.56) of GPTQ-ada is bold, but GPTQ (49.63) is higher. Similar problem in CSR Avg.

2. The improvement of GPTQ with ada is lower than the RTN in Tab. 6 and Tab. 7. Have compared with more baseline methods, and are there situations where there is no improvement?

3. I agree the issue that accelerating the memory I/O is important in some scenes, but I didn’t find related memory evaluation/analysis result. I'd be glad if you could point this to me.

---

> ### Author Response · Authors · 2023-11-20
>
> Dear reviewer ubBB,
> We express our sincere gratitude for the invaluable feedback offered by the reviewer.
>
> ----------------------------
>
> ### **[Weakness 1] The experiment results with activation quantization is expected, and more compared methods such [1] are necessary.**
>
> In response to the reviewers comment, we emphasize that our paper is primarily focused on the algorithmic aspects of weight-only quantization for large language models, particularly in isolating outliers. As we elaborate in Section 3.2 and Appendix A, our method is suitable for weight-only quantization as it is not reliant on specialized INT8 General Matrix Multiply (GEMM) kernels [5], which are constrained by per-OC (Output Channel) grouping.
>
> To provide a comprehensive validation of our approach with additional baselines, we have conducted further experiments concerning both accuracy and latency. For accuracy-related additional experiments, please refer to our response to Question 2 - b.
>
> The findings in [3], [4], [6], and [7] demonstrate that per-OC weight-only group-wise quantization maintains competitive latency compared to cuBLAS [8] baselines, suggesting the practical feasibility of our approach in kernel implementations. In this regard, to further assess the feasibility of per-IC (Input Channel) weight-only quantization, we have implemented a basic inference kernel dedicated to our quantization approach (per-IC). Due to the limited time and resources, we could not fully optimize the per-IC kernel in these experiments. As shown in Table A, our implemented per-IC kernel shows faster latency compared to cuBLAS. Although the per-IC kernel shows slower latency compared to per-OC kernels due to its sub-optimal status, it's noteworthy that our per-IC approach is selectively applied in a layer-wise manner as shown in Figure 6 in the paper. Thus, AdaDim enhances the accuracy performance of per-OC baselines, as detailed in the paper. Consequently, Table A reveals that our method not only improves accuracy over traditional per-OC methods but also shows potential for increased inference latency with specialized per-IC kernels. This highlights the dual benefits of our approach in enhancing both the accuracy and latency of a quantized large language model. We will include detailed implementation information and experimental results in the revised manuscript to address this feedback comprehensively.
>
> Table A. Latency comparison of the first FFN layer on OPT-175B model with 3-bit and 4-bit precision and corresponding kernel selections with hidden dimension size (matrix size, m) is set to 12288 and various group sizes on A100-80GB-GPU.
> | method               | group size    | Weight (4m x m) | Input (m x 1) | Output (4m x 1) | Latency [ms] |
> | -------------------- | ---- | ------ | ----- | ------ | ------------ |
> | cuBLAS               |      | FP16   | FP16  | FP16   | 0.7258       |
> | OPTQ                 | 128  | INT3   | FP16  | FP16   | 0.3599       |
> | AWQ                  | 128  | INT4   | FP16  | FP16   | 0.3238       |
> | LUT-GEMM             | 128  | INT4   | FP16  | FP16   | 0.2688       |
> | LUT-GEMM             | 128  | INT3   | FP16  | FP16   | 0.225        |
> |                      |      |        |       |        |              |
> | per-IC kernel (Ours) | 128  | INT4   | FP16  | FP16   | 0.57701      |
> | per-IC kernel (Ours) | 256  | INT4   | FP16  | FP16   | 0.39219      |
> | per-IC kernel (Ours) | 512  | INT4   | FP16  | FP16   | 0.30298      |
> | per-IC kernel (Ours) | 2048 | INT4   | FP16  | FP16   | 0.27206      |
> |                      |      |        |       |        |              |
> | per-IC kernel (Ours) | 128  | INT3   | FP16  | FP16   | 0.53928      |
> | per-IC kernel (Ours) | 256  | INT3   | FP16  | FP16   | 0.35392      |
> | per-IC kernel (Ours) | 512  | INT3   | FP16  | FP16   | 0.26456      |
> | per-IC kernel (Ours) | 2048 | INT3   | FP16  | FP16   | 0.20333      |
>
> ----------------------------

---

> ### Author Response · Authors · 2023-11-20
>
> ----------------------------
>
> ### **[Weakness 2] The result of A.3 is important, more detailed analysis with diagram is expected.**
>
> In response to the feedback from reviewers, we have measured the end-to-end latency (runtime) of RTN, GPTQ, both with and without AdaDim (Ours), and AWQ as presented in Table B. We utilize LLaMA-2-70B for these experiments on a single A100-80GB GPU. Quantization bit-precision is set to 4-bit with a group size of 128. We are thankful for your comments, which have significantly contributed to the qualitative enhancement of our paper.
>
> Table B. End-to-end latency (Runtime) Comparison with RTN, GPTQ, for both with and without AdaDim (Ours), and AWQ. The table illustrates that while AWQ requires an additional 49 minutes compared to RTN due to its search over scale and clip parameters, AdaDim adds only 5 minutes to the RTN process and an additional 1 minutes to the GPTQ process.
> | Method(w4g128) | End-to-End Latency(min) |
> |:--------------:|:------------:|
> |       AWQ      |      77      |
> |       RTN      |      28      |
> |     RTN-ada    |      33     |
> |      GPTQ      |        91   |
> |    GPTQ-ada    |      92      |
>
> ----------------------------
>
> ### **[Question 1] Typo error: The MMLU Avg. (49.56) of GPTQ-ada is bold, but GPTQ (49.63) is higher. Similar problem in CSR Avg.**
>
> Thank you for pointing out the typo in the appendix section of the manuscript. We will address and correct these typos in the revised version of the manuscript.
>
> ----------------------------
>
> ### **[Question 2 - a] The improvement of GPTQ with ada is lower than the RTN in Tab. 6 and Tab. 7.**
>
> Since RTN is a basic and relatively simplistic uniform quantization method, it experiences notable performance degradation in terms of accuracy, particularly at our main target bit-precision range of 3 bits. In contrast, GPTQ, the current state-of-the-art (SOTA) method for weight-only quantization, outperforms RTN in accuracy. This disparity likely explains why GPTQ's improvement with ada is less pronounced than RTN's when enhanced with ada, as RTN has a greater potential for improvement, evident in Tables 6 and 7 of the paper. This is primarily because RTN begins with a much lower baseline accuracy compared to GPTQ, making it more susceptible to significant gains from ada's ability to isolate outliers in an input-channel manner. Nonetheless, it's crucial to recognize that changing quantization channel dimensions from output-channel to input-channel, regardless of whether RTN or GPTQ is used, leads to an increase in accuracy. We are grateful for the reviewers' valuable feedback, which has contributed to substantial improvements in the manuscript. These revisions will be incorporated into the discussion section of the manuscript.
>
> ----------------------------

---

> ### Author Response · Authors · 2023-11-20
>
> ### **[Question 2 - b] Have compared with more baseline methods, and are there situations where there is no improvement?**
> To expand our comparison with more baseline methods, we conducted additional experiments with SpQR[1, 2], which are novel weight-only quantization methods incorporating small group-wise double quantization and FP16 outlier sparse representation. Table C shows that RTN with ada (RTN-ada) achieves performance comparable to SpQR. This similarity is attributed to RTN with ada adopting a more average bit-precision than SpQR in these experiments. For a more balanced comparison regarding average bit-precision with SpQR, we undertook further experiments at lower bit-precisions, with the results presented in Table D. According to Table D, SpQR surpasses RTN with ada in a similar average bit-precision range. However, it's noteworthy that SpQR requires an FP16 representation for outliers, necessitating an additional sparse inference kernel, which is not a requirement for AdaDim. We would like to emphasize that by shifting the focus of quantization channel dimensions from output-channel to input-channel with AdaDim, the performance of current weight-only group-wise quantization methods can be effectively enhanced.
>
> Table C. Perplexity (PPL) comparison between SpQR [1] and RTN-ada on the Wikitext2 and C4 datasets, employing a 4-bit channel-wise configuration with grouping size of 128. The lower PPL, the better.
> |         |           |     LLaMA-V1-7B     |     LLaMA-V1-33B    |     LLaMA-V1-65B    |
> |:-------:|:---------:|:-------------------:|:-------------------:|:-------------------:|
> |  Method | Avg bits. | Wikitext2 [PPL] / C4 [PPL] | Wikitext2 [PPL] / C4 [PPL] | Wikitext2 [PPL] / C4 [PPL] |
> |   FP16  |     16    |     5.68 / 7.08     |     4.10 / 5.98     |     3.53 . 5.62     |
> | RTN-ada |   4.125   | **5.80** / **7.22** | **4.19** / **6.05** | **3.61** / **5.68** |
> |   SpQR  |    3.9    |     5.87 / 7.28     |     4.25 / 6.08     |     3.68 / 5.70     |
>
> Table D. Perplexity (PPL) comparison between SpQR [1] and RTN-ada on the Wikitext2 and C4 datasets, employing a 3-bit channel-wise configuration with various grouping sizes. The lower PPL, the better.
> | LLaMA-V1-65B | Group-size | Avg Bits | Wikitext2 [PPL] | C4 [PPL] |
> |:------------:|:----------:|:--------:|:----------:|:------:|
> |     FP16     |      -     |    16    |    3.53    |  5.62  |
> |    RTN-ada   |     16     |   4.00   |    3.74    |  5.75  |
> |    RTN-ada   |     32     |   3.50   |    3.82    |  5.80  |
> |    RTN-ada   |     64     |   3.25   |    3.91    |  5.87  |
> |     SpQR     |     16     |   3.63   |    3.74    |  5.73  |
>
> ---------------------
> ### **[Question 2 - c] and are there situations where there is no improvement?**
>
> In light of the memory-wall problem [8] associated with large language models, where memory is the primary bottleneck rather than computation, weight-only quantization methods employ low-bit quantization with grouping techniques. Using smaller group sizes allows these techniques to more effectively isolate outliers in large language models [1]. To identify scenarios where there is no improvement, we conducted experiments without grouping. As indicated in Table E and F, the absence of the grouping technique hinders effective isolation of outliers, resulting in less pronounced improvements with the proposed method, AdaDim.
>
> Table E. Perplexity (PPL) comparison between RTN and RTN-ada on the Wikitext2 and C4 datasets, employing a 4-bit channel-wise configuration without grouping scales. The lower PPL, the better.
> |         | LLaMA-V1-7B         | LLaMA-V1-33B        |
> |---------|---------------------|---------------------|
> | Method  | Wikitext2 [PPL]  / C4 [PPL] | Wikitext2 [PPL] / C4 [PPL] |
> | FP16    | 5.68 / 7.08         | 4.10 / 5.98         |
> | RTN     | 6.29 / 7.73         | 4.54 / 6.33         |
> | RTN-ada | 6.22 / 7.66 | 4.50 / 6.30 |
>
> Table F. Average score comparison between RTN and RTN-ada on the MMLU and Common sense reasoning (CSR) benchmark, employing a 3-bit channel-wise configuration without grouping scales.
> |         |  LLaMA-V2-7B  |  LLaMA-V2-13B |  LLaMA-V1-33B |  LLaMA-V2-70B |
> |:-------:|:-------------:|:-------------:|:-------------:|:-------------:|
> |  Method |   MMLU(%) / CSR(%)  |   MMLU(%) / CSR(%)  |   MMLU(%) / CSR(%)  |   MMLU(%) / CSR(%)  |
> |   FP16  | 45.98 / 67.93 | 55.61 / 70.33 | 58.46 / 72.96 |   69.29 / ?   |
> |   RTN   | 24.57 / 42.62 | 27.11 / 55.82 | 27.13 / 48.12 | 34.56 / 59.72 |
> | RTN-ada | 27.42 / 53.83 | 25.63 / 46.19 | 26.56 / 51.94 | 47.71 / 64.53 |

---

> ### Author Response · Authors · 2023-11-20
>
> ---------------
> ### **[Question 3] I agree the issue that accelerating the memory I/O is important in some scenes, but I didn’t find related memory evaluation/analysis result. I'd be glad if you could point this to me.**
>
> In response to the reviewer's comment, we cite studies [3], [4], [5], and [6]. Our research is primarily focused on modifying the channel dimensions of quantization scales, rather than the integer weights themselves. Consequently, the weight compression effect of our quantization approach aligns with that of other per-OC (Output Channel) weight-only quantization methods. For instance, the FP16 LLaMA-65B model occupies 131 GB of memory. In contrast, both the 4-bit and 3-bit quantized LLaMA-65B models, irrespective of using per-OC or per-IC (Input Channel) quantization, require only 33 GB and 25 GB of memory, respectively, for their model sizes. For a more detailed analysis of memory-wall issues, we kindly refer to [8].
>
> ---------------
> ### **Reference**
>
> [1] Dettmers, Tim, et al. "SpQR: A Sparse-Quantized Representation for Near-Lossless LLM Weight Compression." arXiv preprint arXiv:2306.03078 (2023).
> [2] https://github.com/Vahe1994/SpQR.
> [3] Park, Gunho, et al. "nuQmm: Quantized matmul for efficient inference of large-scale generative language models." arXiv preprint arXiv:2206.09557 (2022).
> [4] Frantar, Elias, et al. "OPTQ: Accurate quantization for generative pre-trained transformers." The Eleventh International Conference on Learning Representations. 2022.
> [5] Jeon, Yongkweon, et al. "Biqgemm: matrix multiplication with lookup table for binary-coding-based quantized dnns." SC20: International Conference for High Performance Computing, Networking, Storage and Analysis. IEEE, 2020.
> [6] Lin, Ji, et al. "AWQ: Activation-aware Weight Quantization for LLM Compression and Acceleration." arXiv preprint arXiv:2306.00978 (2023).
> [7] https://developer.nvidia.com/cublas.
> [8] Kim, Sehoon, et al. "SqueezeLLM: Dense-and-Sparse Quantization." arXiv preprint arXiv:2306.07629 (2023).

---

> ### Comment · Reviewer_ubBB · 2023-11-21
>
> Thank for the authors' response. The authors have partially addressed my concerns. I will raise my score.

---

> > ### Author Response · Authors · 2023-11-21
> >
> > We are pleased to note that some of your concerns have been addressed. Your feedback has been instrumental in this process, and we sincerely extend our gratitude for your invaluable insights. We will ensure that your comments are incorporated into the revised manuscript.

---

### Meta-Review · Area_Chair_JU4L · 2023-12-10

**Metareview:**

This paper proposes a new weight quantization method which creates quantization groups along the input channel axis rather than output channel axis. The proposed method achieves superior performance compared to GPTQ. Reviewers agree that this paper is a solid contribution, contains some interesting insights, and can be practically useful. Weaknesses include limited novelty and some unclear presentation.

**Justification For Why Not Higher Score:**

limited novelty and some unclear presentation

**Justification For Why Not Lower Score:**

this paper is a solid contribution, contains some interesting insights

---

### Decision · Program_Chairs · 2024-01-16

Accept (poster)